# ONLINE CONTINUAL LEARNING WITH FEEDFORWARD ADAPTATION

## ABSTRACT

Recently deep learning has been widely used in time-series prediction tasks. Although a trained deep neural network model typically performs well on the training set, performance drop significantly in a test set under slight distribution shifts. This challenge motivates the adoption of online test-time adaptation algorithms to update the prediction models in real time to improve the prediction performance. Existing online adaptation methods optimize the prediction model by feeding back the latest prediction error computed with respect to the latest observation. However, the feedback based approach is prone to forgetting past information. In this work, we propose an online adaptation method with feedforward compensation, which uses critical data samples from a memory buffer, instead of the latest samples, to optimize the prediction model. We prove that the proposed approach has a smaller error bound than previously used approaches in slow time-varying systems. The experiments on several time-series prediction tasks show that the proposed feedforward adaptation outperforms previous adaptation methods by $12\%$. In addition, the proposed feedforward adaptation method is able to estimate an uncertainty bound of the prediction that is agnostic from specific optimizers, while existing feedback adaptation could not.

## 1 INTRODUCTION

Time-series prediction (or forecasting) has been widely studied in many fields, including control, energy management, and financial investment Box et al. (2015); Brockwell & Davis (2002). Among the research applications, acquiring future trends and tendencies of the time-series data is one of the most important subjects. With the emergence of deep learning, many deep neural network models have been proposed to solve this problem Lim & Zohren (2021), e.g., Recurrent Neural Networks Lai et al. (2018) and Temporal Convolutional Networks Bai et al. (2018). Inspired by the great success of Transformer in the NLP and CV community Vaswani et al. (2017); Dosovitskiy et al. (2020), Transformer-style methods have been introduced to capture long-term dependencies in time series prediction tasks Zhou et al. (2021). Benefiting from the self-attention mechanism, Transformers obtain a great advantage in modeling long-term dependencies for sequential data Brown et al. (2020). Although a trained Transformer model (or other big deep neural network models) typically performs well on the training set, performance can significantly drop in a slightly different test domain or under a slightly different data distribution Popel & Bojar (2018); Si et al. (2019).

In practical time-series prediction problems, there are often significant distributional discrepancies between the offline training set and the real-time testing set. These differences may be attributed to multiple factors. In some cases, it is too expensive to collect large unbiased training datasets, e.g., for weather prediction or medical time-series prediction. In other cases, it may be difficult to obtain the training instances from a specific domain. For example, in human-robot collaboration, it is hard to collect data from all potential future users. In these cases, adaptation techniques are applied to deal with the distribution mismatch between offline training and real-time testing Blum (1998). Besides, some tasks require the system to adapt itself after every observation. For example, in human-robot collaboration, the robot needs to continually adapt its behaviors to different users. In these scenarios, online adaptation techniques are often embraced Abuduweili et al. (2019).

**Online adaptation** is a special case of online continual learning, which continually learns from real-time streaming data. In online adaptation, a prediction model receives sequential observations,

and then an online optimization algorithm (e.g. SGD) updates the prediction model according to the prediction loss measured by the observed data. The goal of online adaptation is to improve prediction accuracy in subsequent rounds. Online adaptation is currently applied to many kinds of research like time-series prediction Pastor et al. (2011); Abuduweili & Liu (2020), image recognition Lee & Kriegman (2005); Chen et al. (2022), and machine translation Martínez-Gómez et al. (2012). In this paper, we mainly focus on time-series prediction tasks, but the proposed methods also can be used for other online adaptation (or online learning) tasks.

Most existing online adaptation approaches are based on feedback compensation Tonioni et al. (2019), analogous to feedback control. In feedback adaptation, a prediction model only utilizes the latest received data. After observing a new sample, the online optimization algorithm updates the prediction model according to the prediction loss measured between the last prediction and the latest ground truth. However, this kind of passive feedback compensation is not efficient.

In this work, we propose feedforward compensation in online adaptation to maximize information extraction from existing data, especially those that are more critical. A critical sample is more helpful to reduce the objective (loss) of the model when the sample is selected for training. In the proposed **feedforward adaptation**, we will not only have forgetting as is done in conventional online adaptation Paleologu et al. (2008), but also enable recalling to compensate for potential short-sighted behaviors due to forgetting. There is a balance between forgetting and recalling. On the one hand, to rapidly learn the new function value in a time-varying system, we need to forget some of the old data. On the other hand, too much forgetting may cause unstable and incorrect predictions when we encounter a similar pattern with historical data. To achieve the balance between forgetting and recalling, we design a novel mechanism for feedforward compensation using a memory buffer similar to the functionality of the Hippocampus in the human brain Barron (2021). We will maintain the memory buffer by storing recent $L$-steps observations (or hidden features) of samples. When the prediction model experiences similar observations, it will pull the corresponding data (critical sample) from the memory buffer to enhance learning. For example, in human behavior prediction tasks, a human subject may exhibit similar behavior patterns on different days. These would be extremely difficult to discover if we only learn from the most recent data like conventional online adaptation but can be identified using the feedforward adaptation methods with memory buffer. We can also use the related information between the current sample and critical samples to measure the uncertainty bound to the current prediction. Our main contributions can be summarized in the following points.

- By summarizing feedforward and feedback adaptation methods, we provide a general online test-time adaptation framework and prove its error bound.
- We propose a feedforward compensation for online test-time adaptation problems. We prove that the proposed feedforward adaptation method has a smaller error bound than previously used feedback methods.
- We propose an uncertainty-bound estimation related to the feedforward approach, which is agnostic from specific optimizers.
- We conduct extensive experiments to show that the proposed feedforward adaptation is superior to conventional feedback adaptation.

## 2 PROBLEM OVERVIEW

### 2.1 TIME-SERIES PREDICTION

The time series prediction problem is to make inferences on future time-series data given the past and current observations. We consider a multi-step prediction problem: using recent $I$ steps' observations to predict future $O$ steps' data. Assume the transition model is composed of a feature extractor (or Encoder) $E$ and a predictor (or decoder) $f$. At time step $t$, the input to the model is $X_t = [x_{t-I+1}, x_{t-I+2}, \cdots, x_t]$, which denotes the stack of $I$-step recent observations. The output of the model is $Y_{t+1} = [y_{t+1}, y_{t+2}, \cdots, y_{t+O}]$, which denotes the stack of $O$-step future predictions. The observations $x_t, y_t$ are vectors that may contain trajectory or feature, and $x_t = y_t$ for some cases (e.g. univariate prediction). The transition model for time series prediction can be formulated as

$$Z_t = E(X_t), \tag{1}$$
$$Y_{t+1} = f_t(Z_t), \tag{2}$$

where $Z_t$ is a hidden feature representation of input $X_t$. Feature extractor $E$ does not change over time, while predictor $f_t$ changes over time. Let $\hat{f}_t$ denote the ground-truth predictor, that generates

ground-truth output $Y_{t+1}$. In online adaptation, we use the parameterized model (e.g. Neural Networks) $\hat{f}(\theta_t, Z_t)$ with learnable parameters $\theta_t$ to estimate the ground-truth predictor $f_t(Z_t)$. This paper assumes that the encoder $E$ is fixed in online adaptation, which could be trained offline before online adaptation or be a non-parametric feature extractor[1].

## 2.2 ONLINE ADAPTATION

Due to the temporal nature of time series prediction, the output (future) space in prediction tasks is not fixed. Since the train set and test set are split in chronological order, train-test distribution mismatch is very common in time-series prediction. Thus, online test-time adaptation is crucial to overcome the distribution mismatch problems and make the prediction robust to time-varying and heterogeneous behaviors.

Online adaptation also can be called *adaptable prediction*, since it makes an inference concurring with updating model parameters. Online adaptation explores local overfitting to minimize the prediction error: at time step $t$, the prediction error $e_{t+1}$ is to be minimized. The optimization objective is shown below

$$\mathcal{L}_{err} = \min_{\theta_t} e_{t+1} = \min_{\theta_t} \|Y_{t+1} - \hat{f}(\theta_t, Z_t)\|_p, \tag{3}$$

where $f_t(Z_t) = Y_{t+1} = [y_{t+1}, y_{t+2}, \cdots, y_{t+O}]$ is the ground truth observation (to be received in the future) and $\hat{f}(\theta_t, Z_t)) = \hat{Y}_{t+1} = [\hat{y}_{t+1}, \hat{y}_{t+2}, \cdots, \hat{y}_{t+O}]$ is the predicted outcome from the learned model parameter $\theta_t$. The adaptation objective can be in any $\ell_p$ norm.

In conventional (feedback) online adaptation, the objective of minimizing the *prediction error* in the future Eq. (3) can be approximated by minimizing the *fitting error* in the past, as shown below

$$\mathcal{L}_{fb} = \min_{\theta_t} \frac{1}{t} \sum_{i=1}^{t} \lambda^{t-i} \|Y_i - \hat{f}(\theta_t, Z_i)\|_p, \tag{4}$$

where $0 < \lambda \leq 1$ is a forgetting factor. The model parameter $\theta_t$ is updated iteratively when new observations are received. Then a new prediction is made using the new estimate. In the next time step, the estimate will be updated again given a new observation, and the process repeats. It is worth noting that the observation we received at time $t$ is $y_t$. The other terms in $Y_t$ remain unknown. This paper focused on adaptation methods using one-step-ahead observation. It is also possible to conduct online adaptation with multi-step ahead observations Abuduweili & Liu (2021).

In this paper, we propose a feedforward adaptation method, whose objective is different from feedback adaptation. In the feedforward adaptation, the objective of minimizing the *prediction error* in the future Eq. (3) can be approximated by minimizing the *upper bound of the prediction error* in the future, as shown below

$$\mathcal{L}_{ff} = \min_{\theta_t} \text{Bound}[e_{t+1}] = \min_{\theta_t} \text{Bound}[\|Y_{t+1} - \hat{f}(\theta_t, Z_t)\|_p]. \tag{5}$$

We will show the algorithms and effectiveness of the feedforward adaptation in section 3. In specific adaptation algorithms, the feedback and feedforward approach differ by sample selection strategy.

---

**Algorithm 1** General Online Adaptation Framework (Adaptable Prediction)

---

**Input:** Initial predictor $f(\theta_0, :)$ with parameters $\theta_0$, Feature Extractor $E$, Optimizer $\mathcal{O}(:, :, :)$
**Output:** Sequence of predictions $\{\hat{Y}_{t+1}\}_{t=1}^{T}$
1: **for** $t = 1, 2, \cdots, T$ **do**
2:     Receive the ground truth observation values $x_t, y_t$
3:     Find the critical input-output pairs $(Z_s, y_{s+1})$ for $1 \leq s < t$, where $Z_s = E(X_s)$
4:     Adaptation: $\theta_t = \mathcal{O}(\theta_{t-1}, \hat{y}_{s+1}, y_{s+1})$, where $\hat{Y}_{s+1} = [\hat{y}_{s+1}, \cdots, \hat{y}_{s+O}] = f(\theta_{t-1}, Z_s)$
5:     Prediction:   $\hat{Y}_{t+1} = [\hat{y}_{t+1}, \cdots, \hat{y}_{t+O}] = f(\theta_t, Z_t)$, where $Z_t = E(X_t)$
6: **end for**

---

[1]In a special case, the feature extractor can be an identity mapping $E(X) = X$, then we adapt the end-to-end neural network model in online adaptation

We provide a general online adaptation framework as shown in algorithm 1, by incorporating conventional feedback with the proposed feedforward adaptation. At time step $t$, after receiving the current observations $(x_t, y_t)$, we select the critical input-output pair $(Z_s, y_{s+1})$ from historical observations. The critical pair was used to adjust the parameters of the prediction model by an online optimizer (e.g. SGD). Then we obtain the current prediction result with the adapted model. The main difference between different adaptation algorithms lies in the critical pair selection strategy (line 3 in algorithm 1). In feedback adaptation, the critical input-output pairs are composed by the latest observations $Z_s = Z_{t-1}$. In random adaptation, the critical input-output pairs are randomly sampled from historical observations $Z_s \sim [Z_1, \cdots, Z_{t-1}]$. We will show in section 3, in feedforward adaptation, the critical input-output pairs are the most similar samples to the current observation $Z_s = \arg\min_Z \|Z_t - Z\|$.

## 3 FEEDFORWARD ADAPTATION APPROACH

### 3.1 ERROR BOUND FOR GENERAL ONLINE ADAPTATION

The theoretical analysis of the paper is based on two basic conditions about the local smoothness property of ground-truth predictors $f_t$.

$K$**-Lipschitz continuity condition**. For a time-step $t$ and $\forall s \in [t - L, t - 1]$, we have local $K$ Lipschitzness for ground-truth prediction function $f_t$ and recent $L$ steps input data:

$$\|f_t(Z_t) - f_t(Z_s)\| \leq K\|Z_t - Z_s\|, \tag{6}$$

where $K$ is the bound (real number) for the change of the value of the function over input space. Intuitively, a Lipschitz continuous function is limited in how fast function value can change over input space. It is proven that every function that has bounded first derivatives is Lipschitz continuous Sohrab (2003). Similar to Eq. (6), we assume our parameterized function $\hat{f}_t(\theta_t, :)$ (e.g. Neural Networks) has Lipschitz continuity with constant value $\hat{K}$:

$$\|\hat{f}(\theta_t, Z_s) - \hat{f}(\theta_t, Z_t)\| \leq \hat{K}\|Z_t - Z_s\|. \tag{7}$$

K-Lipschitz continuity is common in machine learning because neural networks have bounded first derivatives by proper training.

$\delta$ **time-varying condition.** For a time-step $t$ and $\forall s \in [t - L, t - 1]$, assume the ground-truth prediction functions $f_t$ has bounded changes within recent $L$ steps under the same input $Z_s$:

$$\|f_t(Z_s) - f_s(Z_s)\| \leq \delta, \tag{8}$$

where $\delta$ is the bound (real number) for change of the value of the function sequences over time on a fixed input. A bounded $\delta$ time-varying condition is a common condition in time-series tasks because we obtain similar future time-series signals with the same input in most cases.

**Theorem 1 (Error Bound of Online Adaptation)**. Under the $K$-Lipschitz continuity condition (Eq. (6) and (7)) and $\delta$ time-varying condition (Eq. (8)), the (prior) prediction error $e_{t+1}$ of general online adaptation (algorithm 1) has the following upper bound:

$$e_{t+1} \leq K\|Z_t - Z_s\| + \delta + \|Y_{s+1} - \hat{f}(\theta_t, Z_s)\| + \|\hat{f}(\theta_t, Z_s) - \hat{f}(\theta_t, Z_t)\| \tag{9}$$

$$\leq (K + \hat{K})\|Z_t - Z_s\| + \delta + \|Y_{s+1} - \hat{f}(\theta_t, Z_s)\| \tag{10}$$

Please check the proof in appendix C.1.

### 3.2 FEEDFORWARD ADAPTATION

As discussed in section 2.2, the goal of the online adaptation is to minimize the prediction error in the future Eq. (3). Due to the lack of ground-truth value in the current steps, it is not feasible to directly minimize the prediction error. Conventional online adaptation methods approximate the original objective by minimizing the fitting error in the past Eq. (4). In this work, we approximate the original objective by minimizing the bound of the prediction error Eq. (5). Which is equivalent to optimizing the worst-case scenarios. The worst-case performance analysis is really useful in

real-world applications Roughgarden (2021). We will show that, under the following assumption, feedforward adaptation provides better results than feedback adaptation.

**Temporarily Slow Time-varying Assumption.** For the $\delta$ time-varying function $f_t$, assume $\delta$ is time-independent or it slowly changes within recent $L$-steps: $\frac{\partial \delta}{\partial t} \approx 0$.

Note that, we only assume the ground-truth function $f_t$ varying slowly (locally) within recent $L$-steps not (globally) for every step. Thus the assumption is reasonable because the major changes in function value are caused by the change of inputs instead of the time-dependency in many real-world prediction problems. In addition in some cases, $\delta$ is caused by random noises, then which still follow our assumptions. Under the above assumption, we can ignore the term $\delta$ in optimization. Then the optimization objective for the error bound is shown below:

$$\mathcal{L}_{ff} = \min_{\theta_t} \text{Bound}[e_{t+1}] = \min_{\theta_t, s \in [t-L, t-1]} (K + \hat{K}) \|Z_t - Z_s\| + \|Y_{s+1} - \hat{f}(\theta_t, Z_s)\| \quad (11)$$

However, we still can not directly minimize the above objective, because we do not know $K, \hat{K}$. Then a simplification is applied to the above objective to make it applicable. We change the joint minimization over $s$ and $\theta_t$ to a bi-level optimization which first minimizes the first term of objective over sampling time-step $s$, then minimizes the second term of objective over parameter $\theta_t$. Thus, the simplified objective function is shown below:

$$\mathcal{L}_{final} = \min_{\theta_t} \|Y_{s^\star+1} - \hat{f}(\theta_t, Z_{s^\star})\| \quad (12)$$

$$\text{s.t.} \quad s^\star = \underset{s \in [t-L, t-1]}{\arg\min} \|Z_t - Z_s\| \quad (13)$$

In a summary, the proposed feedforward adaptation method selects the most similar samples to the current observation as critical pair $(Z_{s^\star}, y_{s^\star})$ by Eq. (13). Then using the critical pair to optimize the prediction model by Eq. (12).

## 3.3 Uncertainty Estimation

The error bound Eq. (9) provides uncertainty estimation of the prediction results. Here we use estimation of $\tilde{K}_t$ and $\tilde{\delta}$ to approximate real $K$ and $\delta$ in Eq. (9). We use confidence factor $\sigma \in (0, 1]$ to decay the error bound . The uncertainty estimation $\hat{U}_{t+1}$ for prediction $\hat{Y}_{t+1}$ is shown below:

$$\hat{U}_{t+1} = \sigma \cdot \left( \tilde{K}_t \|Z_t - Z_s\| + \tilde{\delta} + \|Y_{s+1} - \hat{f}(\theta_t, Z_s)\| + \|\hat{f}(\theta_t, Z_s) - \hat{f}(\theta_t, Z_t)\| \right) \quad (14)$$

Where $s = t - 1$ for feedback adaptation and $s = s^\star$ (Eq. (13)) for feedforward adaptation. The confidence factor $\sigma$ is a predefined hyperparameter, e.g. $\sigma = 0.9$ for 90% confidence of uncertainty. $\tilde{\delta}$ and $\tilde{K}_t$ are also predefined hyperparameters, e.g. $\tilde{\delta} = 0.001, \tilde{K} = 1$. For more accurate uncertainty estimation, we can set $\tilde{\delta}$ as a small fixed value, and iteratively update $\tilde{K}_t$ according to the estimated uncertainty $\hat{U}_t$ and real error $e_t$. If the previous uncertainty estimation is much larger than the real error then we shrink the $\tilde{K}_t$ value. If the previous uncertainty estimation is much smaller than the real error then we enlarge the $\tilde{K}_t$ value. The $\tilde{K}_t$ estimation criteria are shown below.

$$\tilde{K}_t = \frac{\tilde{K}_{t-1}}{1.5} \text{ if } \hat{U}_t \geq 2e_t, \ \tilde{K}_t = 1.5\tilde{K}_{t-1} \text{ if } \hat{U}_t \leq 0.5e_t, \ \tilde{K}_t = \tilde{K}_{t-1} \text{ if } 0.5e_t < \hat{U}_t < 2e_t \quad (15)$$

In adaptation and prediction, feedforward adaptation does not require $K$ and $\delta$ values. In the uncertainty estimation, we need to approximate $K$ and $\delta$ or serve these as predefined hyperparameters, but an accurate approximation is not necessary. Since we mainly consider the relative uncertainty estimation between different samples or different optimizers. As shown in Eq. (14), the inaccuracy of $\tilde{K}$ and $\tilde{\delta}$ mainly raise a data-independent error, which is no impact on the data-selection strategy and optimizers. The overall feedforward adaptation algorithm is shown in algorithm 2. The $1 \sim 5$ lines of algorithm 2 is corresponding to the $1 \sim 5$ lines of algorithm 1. Buffer $B$ is used to store recent $L$-step observations.

---

**Algorithm 2** Online Adaptation with Feedforward Compensation

---

**Input:** Initial predictor $f(\theta_0, :)$ with parameters $\theta_0$, Feature Extractor $E$, Optimizer $\mathcal{O}(:, :, :)$
**Input:** Empty $L$-size buffer $B$ ;
**Output:** Sequence of predictions $\{\hat{Y}_{t+1}\}_{t=1}^T$ and estimated uncertainty $\{\hat{U}_{t+1}\}_{t=1}^T$
  1: **for** $t = 1, 2, \cdots, T$ **do**
  2:    Receive the ground truth observation values $x_t, y_t$; Construct input $X_t = [x_{t-I+1}, \cdots, x_t]$
  3:    Find the critical (similar) input-output pairs $(Z_{s^\star}, y_{s^\star+1})$ from buffer $B$ by Eq. (13)
  4:    Adaptation by Eq. (12): $\theta_t = \mathcal{O}(\tilde{\theta}_t, \hat{y}_{s^\star+1}, y_{s^\star+1})$, where $[\hat{y}_{s^\star+1}, \cdots, \hat{y}_{s^\star+O}] = f(\tilde{\theta}_t, Z_{s^\star})$
  5:    Prediction: $\hat{Y}_{t+1} = [\hat{y}_{t+1}, \cdots, \hat{y}_{t+O}] = f(\theta_t, Z_t)$, where $Z_t = E(X_t)$
  6:    Uncertainty $\hat{U}_{t+1}$ Estimation by Eq. (14) and Eq. (15)
  7:    Add current data to buffer: $B.append(Z_t, y_t)$
  8:    **if** $\text{size}(B) > L$ **then**
  9:       $B \leftarrow \text{keep\_more\_recent\_samples}(B, L)$
 10:    **end if**
 11: **end for**

---

### 3.4 Effectiveness of Feedforward Adaptation

As described in section 3.2, under the temporarily slow time-varying assumption, feedforward adaptation achieves a smaller error bound than feedback adaptation. In this section, we analyze more theoretical results on the online adaptation of some specific functions.

**Linear time-variant function**. Consider a function $f_t$ which can be separated into time-invariant function $g(z_t)$ and linearly time-dependent part $\alpha t$. Input $z_t$ is a random variable from the uniform distribution $\mathcal{U}(0, 1)$.

$$f_t(z_t) = g(z_t) + \alpha t, \quad z_t \sim \mathcal{U}(0, 1) \tag{16}$$

**Lemma 1**. We use neural network $f(\hat{\theta}, :)$ to learn the linear time-variant function Eq. (16). Let $K$ be a Lipschitz constant for the ground-truth function $f_t$, and $\hat{K}$ is a Lipschitz constant for neural network $f(\hat{\theta}, :)$. Let $B_e^{fb}$ and $B_e^{ff}$ denote the prediction error bound for feedback and feedforward adaptation. Then we have the following results.

(a) Expectation of the error bound for feedback adaptation is $\mathbb{E}[B_e^{fb}] = \frac{K+\hat{K}}{3} + \alpha$.

(b) Expectation of the error bound for feedforward adaptation is
$\mathbb{E}[B_e^{ff}] \leq (K + \hat{K}) \cdot \max(\frac{1}{t}, \frac{1}{L+1}) + \alpha \cdot \min(\frac{t}{2}, \frac{L+1}{2})$. With $L = \sqrt{\frac{2(K+\hat{K})}{\alpha}} - 1$, we achieve the smallest final error bound for feedforward adaptation: $\mathbb{E}[B_e^{ff}] \leq \sqrt{2\alpha(K + \hat{K})}$.

(c) If time-varying factor $\alpha$ is smaller, specifically $\alpha \leq \alpha^\star = \frac{4-\sqrt{12}}{6}(K + \hat{K})$, feedforward adaptation has smaller error bound than feedback adaptation.

**Time-invariant polynomial function**. Consider a time-invariant polynomial ground-truth function $f_t(z_t)$ with the input of random variable sampled from the uniform distribution $\mathcal{U}(0, 1)$.

$$y_{t+1} = f_t(z_t) = \sum_{i=1}^N W_i z_t^i, \quad z_t \sim \mathcal{U}(0, 1) \tag{17}$$

**Lemma 2**. We use linear projection $\hat{f}(V_t, z_t) = V_t z_t$ to learn the time-invariant polynomial function Eq. (17). Then we have the following results.

(a) Expectation of the prediction error for feedback adaptation is $\mathbb{E}[e_{t+1}^{fb}] \geq \frac{1}{3} \frac{N-1}{N^2} \sum_{i=1}^{N-1} W_{i+1}$.

(b) Expectation of the prediction error for feedforward adaptation is
$\mathbb{E}[e_{t+1}^{ff}] \leq \frac{1}{2} \max(\frac{1}{t}, \frac{1}{L+1}) \sum_{i=1}^{N-1} W_{i+1}$. The final error converges $\lim_{L\to\infty, t\to\infty} \mathbb{E}(e_{t+1}^{ff}) = 0$.

(c) Feedforward adaptation is provably better than feedback adaptation.

The proof of Lemma 1 and Lemma 2 are provided in appendix C.2. Thus feedforward adaptation can be used to learn time-invariant functions and slow time-varying systems. These kinds of systems are very common in the world. For example, the exchange rate and incidence rate for common diseases have a slow time-dependent shift over time.

## 4 EXPERIMENTS

### 4.1 SYNTHETIC EXPERIMENTS

To evaluate Lemma 1 in section 3.4, we consider the following time-varying function.

$$y_{t+1} = f_t(z_t) = \sin z_t + \alpha t, \quad z_t \sim \mathcal{U}(0, 1) \tag{18}$$

According to Lemma 1, we derive the error bound for feedback adaptation $B_e^{fb}$, and feedforward adaptation $B_e^{ff}$ is shown below (please check details in appendix D.1).

$$\mathbb{E}[B_e^{fb}] = \frac{5}{12} + \alpha, \quad \mathbb{E}[B_e^{ff}] \leq \frac{5}{4(L+1)} + \alpha \frac{L+1}{2} \tag{19}$$

$$\mathbb{E}[B_e^{ff}] \leq \sqrt{\frac{5}{2}\alpha} \text{ with } L = \sqrt{\frac{5}{2\alpha}} - 1 \tag{20}$$

Then we calculate the threshold $\alpha^\star$. If $\alpha \leq \alpha^\star$, feedforward adaptation has a smaller error bound.

$$\alpha^\star \approx 0.1, \quad L^\star = \sqrt{\frac{5}{2\alpha^\star}} - 1 \approx 3 \tag{21}$$

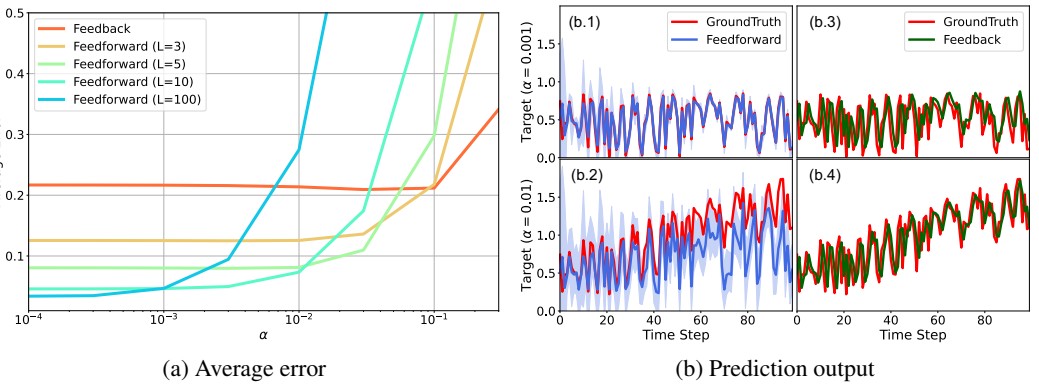

(a) Average error            (b) Prediction output

Figure 1: (a) Average error of feedback and feedforward adaptation with different buffer size $L$ and time-varying factor $\alpha$. (b) Prediction output of feedforward (blue curve) and feedback adaptation (green curve) with buffer size $L = 100$ and time-varying factor $\alpha = 10^{-3}$ and $\alpha = 10^{-2}$.

Figure 1a shows the average prediction error of feedback and feedforward adaptation under different time-varying factors $\alpha$. Note that, feedforward adaptation with $L = 1$ is the same as feedback adaptation. We will show that the results in Fig. 1a correspond to the theoretical results from Lemma 1. (1) As can be seen, for a smaller time-varying factor $\alpha$, feedforward adaptation with a larger buffer size $L$ achieves a smaller prediction error. Besides, if $\alpha$ is larger, the performance of the feedforward adaptation drops. These are consistent with Eq. (20). For example, feedforward adaptation with buffer size $L = 100$ achieves the smallest prediction error when $\alpha < 10^{-3}$. (2) If $\alpha < 0.1$, feedforward adaptation with buffer size $L = 3$ performs better than feedback adaptation. But if $\alpha > 0.1$, feedback adaptation is better. The threshold $\alpha^\star \approx 0.1$ is consistent with Eq. (21).

Figure 1b illustrate the comparison between feedforward and feedback adaptation. The first row shows the prediction results for $\alpha = 10^{-3}$ target function of feedforward adaptation with $L = 100$ (b.1 subfigure) and feedback adaptation (b.3 subfigure). As can be seen, feedforward adaptation can learn the ground truth more precisely. The second row shows the prediction results for $\alpha = 10^{-2}$ target function. For this case, prediction with feedforward adaptation (b.2 subfigure) has a significant shift from the ground truth, although it is still able to learn some input-dependent details. Compared to feedback adaptation, feedforward adaptation is more focused on the input-dependent part of the ground truth function and pays less attention to the time-varying shift of the function. These empirical results validate the conclusions in section 3.4. The experimental evaluation of Lemma 2 is shown in appendix D.2.

## 4.2 EXPERIMENTS ON REAL WORLD DATA

In this section, we evaluate the proposed feedforward adaptation on four real-world benchmarks, including ETT (Electricity Transformer Temperature), Exchange-Rate, ILI (Influenza-like Illness), and THOR human motion trajectory dataset. We include two models as our learnable prediction function $\hat{f}(\theta, :)$: Informer Zhou et al. (2021) and simple MLP (multi-layer perception).

**Experimental Design**. First we offline train models (Informer and MLP) on a train set. After training, we will incrementally receive the data point from the test set like in real-world applications. At each time step, we conduct an online adaptation to optimize the model with selected previous observations by feedback or feedforward compensation. Then the prediction output is inferred from the updated model. We evaluate the prediction results with mean squared error (MSE) and mean absolute error (MAE). In the experiments, we only adapt the decoder of the model and make the encoder fixed. Please check the detailed experimental design in appendix D.3.

**Baselines**. We compare the proposed method with four baselines. 1) *w/o adapt* directly conduct prediction without adaptation. Which is a lower bound for adaptation methods. 2) *Feedback adaptation* is the most important baseline to us. 3) *Random adaptation* is a method that selects the critical pair from the $L$-size buffer with random sampling. 4) *Full adaptation* is a method that uses all samples from the buffer to adapt the model, which is similar to offline training.

**Results**. The detailed results of the experiment on SGD optimizer are shown in table 1. In table 1, the last row denotes the performance gain of the proposed feedforward adaptation over feedback adaptation in terms of average result. Feedforward adaptation achieves the best results on all four datasets. Specifically, feedforward adaptation outperforms feedback adaptation by $12.6\%, 7.4\%, 22.34\%$, and $8.1\%$ in terms of average MSE on four datasets respectively. Feedforward adaptation is better than full adaptation, which means adapting models with more samples is not effective, because online adaptation focuses on the ability to rapidly learn and adapt in the presence of non-stationarity instead of generalization ability like offline learning. Feedforward adaptation is better than feedback and random adaptation, which means the sample selection strategy based on sample similarity in feedforward adaptation is more critical than the time-based sample selection and random sampling.

Table 1: Performance comparison between the proposed feedforward adaptation method and other baselines. Avg denotes the average results of two models (MLP and Informer). The last row denotes the performance gain of feedforward adaptation over feedback adaptation. We use **boldface** and underline for the best and second-best average results.

| Method \ Dataset | | ETTh1 | | Exchange | | ILI | | THOR | |
|---|---|---|---|---|---|---|---|---|---|
| Adaptation | Model | MSE | MAE | MSE | MAE | MSE | MAE | MSE | MAE |
| w/o Adaptation | MLP | 0.195 | 0.371 | 0.549 | 0.540 | 4.348 | 1.413 | 0.135 | 0.117 |
| | Informer | 0.211 | 0.389 | 1.128 | 0.858 | 4.942 | 1.531 | 0.137 | 0.171 |
| | Avg | 0.203 | 0.380 | 0.839 | 0.699 | 4.645 | 1.472 | 0.136 | 0.144 |
| Full Adaptation | MLP | 0.142 | 0.311 | 0.362 | 0.443 | 3.844 | 1.340 | 0.127 | 0.150 |
| | Informer | 0.146 | 0.307 | 0.503 | 0.567 | 3.413 | 1.252 | 0.213 | 0.287 |
| | Avg | 0.144 | 0.309 | 0.433 | 0.505 | 3.628 | 1.296 | 0.170 | 0.219 |
| Random Adaptation | MLP | 0.145 | 0.312 | 0.365 | 0.445 | 3.865 | 1.347 | 0.132 | 0.148 |
| | Informer | 0.131 | 0.283 | 0.497 | 0.553 | 3.867 | 1.380 | 0.204 | 0.271 |
| | Avg | 0.138 | 0.298 | 0.431 | 0.499 | 3.866 | 1.364 | 0.168 | 0.210 |
| Feedback Adaptation | MLP | 0.153 | 0.317 | 0.349 | 0.442 | 3.868 | 1.334 | 0.112 | 0.119 |
| | Informer | 0.116 | 0.269 | 0.326 | 0.461 | 4.383 | 1.411 | 0.134 | 0.173 |
| | Avg | 0.135 | 0.293 | 0.338 | 0.452 | 4.123 | 1.373 | 0.123 | 0.146 |
| Feed Forward Adaptation | MLP | 0.128 | 0.286 | 0.347 | 0.432 | 3.041 | 1.193 | 0.102 | 0.108 |
| | Informer | 0.109 | 0.259 | 0.276 | 0.399 | 3.363 | 1.312 | 0.125 | 0.156 |
| | Avg | **0.118** | **0.272** | **0.311** | **0.415** | **3.202** | **1.252** | **0.113** | **0.132** |
| Gain of Feedforward Over Feedback | | 12.6% | 7.2% | 8.0% | 8.2% | 22.3% | 16.1% | 8.1% | 9.6% |

We found another interesting result by comparison between Informer and MLP on table 1. Without adaptation, the simple MLP outperforms the Informer on average. This is caused by the overfitting of the Informer on the training set. However, by online adaptation, especially with feedforward adaptation, the performance of Informer is greatly increased. For example, the performance of Informer on the ETTh1 dataset increased from 0.211 to 0.109 by 48% and outperform the MLP model. That shows Informer has greater representation ability but may perform poorly on the test

set. Considering this case, it is embarrassing to directly use the big Transformer-like models on real-world time-series prediction. But online adaptation may make the big Transformer-like models powerful again. This phenomenon emphasizes the importance of online adaptation in real-world time-series prediction.

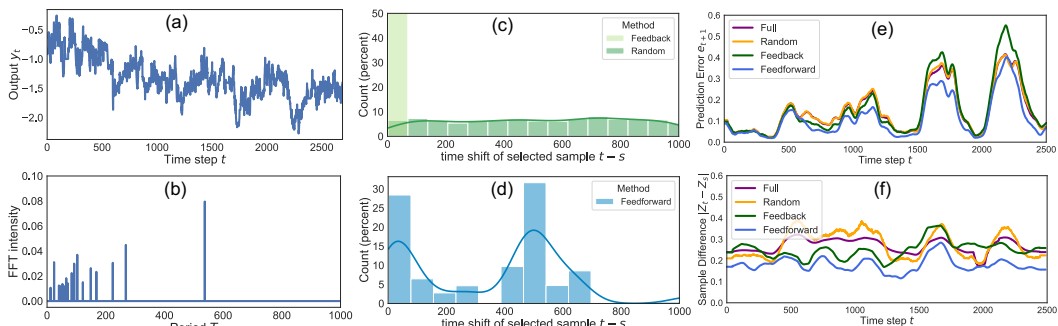

Figure 2: Experiments on ETTh1 data. (a) Time series output; (b) FFT period analysis; (c) Time shift $t - s$ between current sample $Z_t$ and selected sample $Z_s$ in feedback and random adaptation; (d) Time shift $t - s$ in feedforward adaptation; (e) Online prediction error. (f) Sample difference $\|Z_t - Z_s\|$ between current sample $Z_t$ and adapted sample $Z_s$.

**Discussion**. We will show that the sample selection strategy in the proposed feedforward adaptation method could intrinsically mine the periodicity of the input data. Figure 2(a) shows the ETTh1 time-series data, and Fig. 2(b) is the FFT (Fast Fourier Transformation) period analysis of the ETTh1 data. As can be seen ETTh1 has roughly $T \approx 500$ repetition periods. Random adaptation randomly selects samples from the buffer, then the time shift between current time-step $t$ and the time-step $s$ for selected samples is $(t - s) \sim [1, L]$; Feedback adaptation only selects the last time sample to optimization, then $(t - s) = 1$. This can be found from the distribution of time shift $(t - s)$ in Fig. 2(c). For the proposed feedforward adaptation in Fig. 2(d), many samples were selected from $(t - s) \approx 500$ steps earlier, which corresponds to the repetition period of $T \approx 500$. Because feedforward adaptation selects the most similar samples to the current sample, which helps to extract the hidden periodicity of the input signal over time. Thus the distribution of $t - s$ is similar to FFT period analysis. Experiments on different datasets are shown in appendix D.5. According to theorem 1 (Eq. (10)), the error bound is related to the sample difference $\|Z_t - Z_s\|$. Figure 2(e) shows the real prediction error for different adaptation methods over time. As can be seen, feedforward adaptation has the smallest prediction error, because feedforward adaptation has the smallest sample difference $\|Z_t - Z_s\|$ during adaptation, as shown in Fig. 2(f).

One of the advantages of the proposed feedforward adaptation is that it could provide uncertainty estimation. The results of prediction output and uncertainty are shown in appendix D.6.

## 5 CONCLUSION

This paper studies an effective feedforward adaptation algorithm for time-series prediction tasks. Firstly we propose the general framework for online adaptation which includes feedback and feedforward adaptation. Then we propose the feedforward adaptation algorithm by selecting the most similar critical samples for optimization. We prove that, in a time-invariant or slow time-varying system, the feedforward adaptation has a smaller error bound than conventional feedback adaptation. In the experiments, we empirically validate the effectiveness of the proposed algorithm both on synthetic and real-world data. In the end, we show that online adaptation can greatly improve the performance of Informer or other models.

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

## A    RELATED WORKS

**Time series prediction.** Time-series prediction historically has very diverse applications across various domains Lim & Zohren (2021). These applications trigger many methods for time-series prediction. Modern machine learning methods provide a purely data-driven manner without domain knowledge Ahmed et al. (2010). In recent years, deep neural networks becomes the dominant approach inspired by notable achievements of deep learning in computer vision Dosovitskiy et al. (2020) and natural language processing Vaswani et al. (2017). Among deep neural networks, Transformer-style models are becoming more and more popular in time-series prediction tasks, since which has the potential to capture long-term dependencies Brown et al. (2020). Most Transformer-style models focus on long-term time series forecasting problem, including Informer Zhou et al. (2021), Autoformer Wu et al. (2021), and Pyraformer Liu et al. (2021).

**Continual Learning** also known as Lifelong learning, aims at providing incrementally updated knowledge in an ever-changing environment. From the data statistics perspective, continual learning studies the problem of learning from non-i.i.d data, with the goal of preserving and extending the acquired knowledge. Conventional continual learning assumes the new data arrive one task at a time, and the data distribution for each task is stationary De Lange et al. (2021). Which is also called offline continual learning. Different from the batch training on offline continual learning, **Online Continual Learning** focuses on the more realistic problem, where data arrive one tiny batch at a time and previously seen batches from the current or the previous tasks are not accessible Mai et al. (2022). Therefore, online continual learning is required to efficiently learn from a single sample from the online data stream in a single step. Recent works of continual learning can be roughly divided into four families: dynamic architectures, regularization-based, rehearsal, and generative replay approach Lesort et al. (2020). Among these methods, Rehearsal-based methods are more related to our work, which use memory replays to enhance the knowledge from the previous data or processes such as iCaRL Rebuffi et al. (2017), GEM Lopez-Paz & Ranzato (2017). However, most of these methods focus on offline continual learning.

**Test-time Adaptation** is a special setting of domain adaptation and continual learning where a trained model on the training domain has to adapt to the testing domain without accessing the training set Kundu et al. (2020). Most test-time adaptation methods for image data utilize unsupervised domain adaptation algorithms to improve the performance on the test set such as entropy minimization Wang et al. (2020), pseudo-labeling Li et al. (2020). For the time-series prediction task, the input signal at the current step might be the ground-truth label of the previous step. So (partially) supervised learning can be used for test-time adaptation on time-series tasks Abuduweili & Liu (2021). In many real-world applications, online test time adaptation (online adaptation for short) is more reasonable than offline adaptation. For example, in automotive driving, we only receive an observation (sample) in an online manner at each time step, instead of batch data of offline setting.

**Online Adaptation** continually learns from real-time streaming data. Online adaptation is a special case of online continual learning in that the model is always offline trained on a training dataset (which can be very small), and then incrementally adapt its models in an online test dataset Cheng et al. (2019). Unlike works that focus on avoiding catastrophic forgetting in offline continual learning Kirkpatrick et al. (2017), online adaptation focuses on the ability to rapidly learn and adapt in the presence of non-stationarity. Most existing online adaptation approaches are based on feedback compensation with SGD (Stochastic Gradient Descent) or EKF (Extended Kalman Filter) optimizers Bhasin et al. (2012).

## B    ONLINE OPTIMIZERS

In algorithm 1, an online optimizer was used to adjust the model with the selected critical samples: $\theta_t = \mathcal{O}(\theta_{t-1}, \hat{y}_{s+1}, y_{s+1})$. Unlike offline machine learning, online optimizers need to handle iterative input-output pairs with a single sample. We assume our objective function (loss function) has suitable smoothness properties (e.g. differentiable and locally Lipschitz continuous). SGD and EKF are two widely used optimizers for online adaptation.

**Stochastic Gradient Descent (SGD)** is one of the widely used optimizers in online adaptation Kivinen et al. (2004). SGD is memory efficient and stable. The update formula of the SGD is shown

below:

$$\theta_t = \theta_{t-1} - \eta \left( \frac{\partial}{\partial \theta} \|y_{s+1} - \hat{f}(\theta, Z_s)\|^2 \right) |_{\theta=\theta_{t-1}}, \tag{22}$$

where $\eta$ is the learning rate.

**Extended Kalman Filter (EKF)** is another strong optimizer in online adaptation Jazwinski (2007). In the original EKF, the object being estimated is the state value of a dynamic system. In online adaptation, we can apply the EKF approach to adapt model parameters by regarding model parameters as system states. The EKF approach has a faster convergence rate than SGD, but it is very memory-exhaustive. When optimizing small neural networks, EKF approach has been demonstrated to be superior to the SGD-based algorithms Ruck et al. (1992); Abuduweili & Liu (2021). If the adapted model size is not big, EKF is a strong optimizer for online adaptation. The update formula of the EKF is shown below:

$$H_t = \frac{\partial \hat{f}(\theta, Z_s)}{\partial \theta} |_{\theta=\theta_{t-1}} \tag{23}$$

$$G_t = P_{t-1} \cdot H_t^{\mathsf{T}} \cdot (H_t \cdot P_{t-1} \cdot H_t^{\mathsf{T}} + \lambda I)^{-1} \tag{24}$$

$$\theta_t = \theta_{t-1} + G_t \cdot (y_{s+1} - \hat{f}(\theta_{t-1}, Z_s)) \tag{25}$$

$$P_t = \lambda^{-1}(P_{t-1} - G_t \cdot H_t \cdot P_{t-1}) \tag{26}$$

where $\lambda \in (0, 1]$ is a forgetting factor, $H_t$ is the gradient matrix, $G_t$ is the Kalman gain, and $P_t$ is a matrix representing the uncertainty in the estimates of parameters. The initial value of the matrix $P_t$ can be set as a diagonal matrix $P_0 = p_0 \cdot \mathrm{diag}(1, 1, \cdots, 1)$ for $p_0 > 0$.

## C  THEORETICAL RESULTS ON SECTION 3

### C.1  THEOREM 1 (ERROR BOUND OF ONLINE ADAPTATION)

**Bound of ground-truth difference**. If the sereis of ground-truth functions $f_t$ within recent $L$ steps follows $K$-Lipschitz continuity Eq. (6) and $\delta$ time-varying conditions Eq. (8), then the ground-truth value $Y_{t+1}$ and $Y_{s+1}$ has the following property:

$$\|Y_{t+1} - Y_{s+1}\| \leq K\|Z_t - Z_s\| + \delta \tag{27}$$

The proof is shown below:

$$
\begin{aligned}
\|Y_{t+1} - Y_{s+1}\| &= \|f_t(Z_t) - f_s(Z_s)\| \\
&= \|f_t(Z_t) - f_t(Z_s) + f_t(Z_s) - f_s(Z_s)\| \\
&\leq \|f_t(Z_t) - f_t(Z_s)\| + \|f_t(Z_s) - f_s(Z_s)\| \quad \text{(triangle inequality)} \\
&\leq K\|Z_t - Z_s\| + \|f_t(Z_s) - f_s(Z_s)\| \quad \text{(K Lipschitzness)} \\
&\leq K\|Z_t - Z_s\| + \delta \quad (\delta \text{ time varying})
\end{aligned} \tag{28}
$$

**Error Bound of Online Adaptation**. For time step $t$, the (prior) prediction error $e_{t+1}$ has the following inequality:

$$
\begin{aligned}
e_{t+1} = \|Y_{t+1} - \hat{Y}_{t+1}\| &= \|Y_{t+1} - \hat{f}(\theta_t, Z_t)\| \\
&= \|Y_{t+1} - Y_{s+1} + Y_{s+1} - \hat{f}(\theta_t, Z_s) + \hat{f}(\theta_t, Z_s) - \hat{f}(\theta_t, Z_t)\| \\
&\leq \|Y_{t+1} - Y_{s+1}\| + \|Y_{s+1} - \hat{f}(\theta_t, Z_s)\| + \|\hat{f}(\theta_t, Z_s) - \hat{f}(\theta_t, Z_t)\| \quad \text{(triangle inequality)} \\
&\leq K\|Z_t - Z_s\| + \delta + \|Y_{s+1} - \hat{f}(\theta_t, Z_s)\| + \|\hat{f}(\theta_t, Z_s) - \hat{f}(\theta_t, Z_t)\| \quad \text{(Eq. (27))}
\end{aligned} \tag{29}
$$

The first two terms come from the difference between ground-truth $Y_{t+1} - Y_{s+1}$, the third term is a (posterior) fitting error for input-output tuple $(Z_s, Y_{s+1})$, and the last term is the difference between two predictions. Combining Eq. (29) with Eq. (7), we obtain the error bound for general online adaptation is shown below:

$$e_{t+1} \leq (K + \hat{K})\|Z_t - Z_s\| + \delta + \|Y_{s+1} - \hat{f}(\theta_t, Z_s)\| \tag{30}$$

## C.2 EFFECTIVENESS OF FEEDFORWARD ADAPTATION

### C.2.1 ERROR BOUND OF ONLINE ADAPTATION ON LINEAR TIME-VARIANT FUNCTION

In this section, we compare the error bound of feedforward and feedback adaptation methods on linear time-variant systems (functions). Consider a function $f_t$ which can be separated into time-invariant function $g(z_t)$ and linearly time-dependent part $\alpha t$. Input $z_t$ is a random variable from the uniform distribution $\mathcal{U}(0, 1)$.

$$z_t \sim \mathcal{U}(0, 1) \tag{31}$$

$$f_t(z_t) = g(z_t) + \alpha t \tag{32}$$

We use neural network $f(\hat{\theta}, :)$ to learn the above function. Let $K$ be a Lipschitz constant for $f_t$ (it is also equal to the Lipschitz constant for the function $g$), and $\hat{K}$ is a Lipschitz constant for neural network $f(\hat{\theta}, :)$. The $\delta$ time-varying condition becomes $\|f_t(z_s) - f_s(z_s)\| = \alpha\|t - s\|$, and time-varying factor $\delta = \alpha\|t - s\|$. Then we have an error bound from Eq. (10):

$$e_{t+1} \leq (K + \hat{K})\|z_t - z_s\| + \alpha\|t - s\| + \|y_{s+1} - \hat{f}(\theta_t, z_s)\| \tag{33}$$

Now we consider the comparison of the error bound between feedforward and feedback adaptation. Note that, the last term of the above equation $\|y_{s+1} - \hat{f}(\theta_t, z_s)\|$ is a fitting error on input-output tuple $(z_s, y_{s+1})$, and the fitting error is irrelevant to feedback or feedforward compensation strategy. Besides, in over-parameterized neural networks, the fitting error is very small, even can be zero Allen-Zhu et al. (2019). Thus we ignore the fitting error in comparison. We now compare the expectation of the error bound. Let $B_e$ denote the prediction error bound for online adaptation, $B_e := \text{Bound}[e_{t+1}]$. Then we have the expectation of the error bound for online adaptation as shown below.

$$\mathbb{E}[B_e] = (K + \hat{K})\mathbb{E}[\|z_t - z_s\|] + \alpha\mathbb{E}[|t - s|] \tag{34}$$

Then we use Eq. (34) to derive the expectation of the error bound for feedback and feedforward adaptation methods. The core of the proof is to estimate $\mathbb{E}[\|z_t - z_s\|]$ and $\mathbb{E}[|t - s|]$.

In the following sections, we compare the error bounds of four methods. 1) *Feedback adaptation* is the most important baseline that selects the latest observations to optimize the model. 2) *Random adaptation* is a method that randomly selects samples from the $L$-size buffer to optimize the model. 3) *Full adaptation* is a method that uses all samples from the buffer to adapt the model, which is similar to offline training. 4) *Feedforward adaptation* is the proposed method, that selects the most similar samples to optimize the model.

**Expectation of the error bound for feedback adaptation**. In feedback adaptation, the selected input-output pairs are the latest observations $z_s = z_{t-1}$ and $s = t - 1$. The current sample $z_t$ and last sample $z_{t-1}$ are independent random variables from $\mathcal{U}(0, 1)$. The expectation of the distance between these two independent variables is $\frac{1}{3}$, then $\mathbb{E}[\|z_t - z_s\|] = \mathbb{E}[\|z_t - z_{t-1}\|] = \frac{1}{3}$. We have the expectation for the error bound of feedback adaptation $B_e^{fb}$ as shown:

$$\mathbb{E}[B_e^{fb}] = (K + \hat{K})\mathbb{E}[\|z_t - z_s\|] + \alpha\mathbb{E}[|t - s|]$$

$$= (K + \hat{K}) \cdot \frac{1}{3} + \alpha \cdot 1 = \frac{K + \hat{K}}{3} + \alpha \tag{35}$$

**Expectation of the error bound for feedforward adaptation**. In feedforward adaptation, the selected input-output pairs are the most similar samples to the current observation $z_s^\star = \arg\min_{z_s} \|z_t - z_s\|$ from $L$-size buffer, and $s^\star = \arg\min_{s \in [t-L, t-1]} \|Z_t - Z_s\|$. The expectation $\mathbb{E}[\|z_t - z_s\|]$ represents the average minimum distance between current sample $z_t$ and previous samples. If $t \leq L$, $\mathbb{E}[\|z_t - z_s\|]$ considers the minimum distance for $t$ samples, which is no greater than $\frac{1}{t}$. If $t > L$, $\mathbb{E}[\|z_t - z_s\|]$ considers the minimum distance for $L + 1$ samples in the buffer, which is no greater than $\frac{1}{L+1}$. Similarly, $\mathbb{E}[\|t - s\|]$ is an average distance between $t$ indices for $t \leq L$, which is $\frac{t}{2}$, or average distance from $L + 1$ indices for $t \geq l$, which is $\frac{L+1}{2}$. We have the expectation for the error bound of feedforward adaptation $B_e^{ff}$ as shown:

$$\mathbb{E}[B_e^{ff}] = (K + \hat{K})\mathbb{E}[\|z_t - z_s\|] + \alpha\mathbb{E}[|t - s|]$$

$$\leq (K + \hat{K}) \cdot \max(\frac{1}{t}, \frac{1}{L+1}) + \alpha \cdot \min(\frac{t}{2}, \frac{L+1}{2}) \tag{36}$$

We mainly consider the final performance $t > L$, then the above equation can be simplified as:

$$\mathbb{E}[B_e^{ff}] \leq \frac{K + \hat{K}}{L + 1} + \alpha \frac{L + 1}{2} \tag{37}$$

With $L = \sqrt{\frac{2(K+\hat{K})}{\alpha}} - 1$, we achieve the smallest error bound for feedforward adaptation:

$$\mathbb{E}[B_e^{ff}] \leq \sqrt{2\alpha(K + \hat{K})} \tag{38}$$

**Expectation of the error bound for random adaptation**. In random adaptation, the critical input-output pairs are randomly sampled from historical observations $z_s \sim [z_{t-L}, \cdots, z_{t-1}]$ from $L$ size buffer. The current input $z_t$ and selected input $z_s$ are independent random variables from $\mathcal{U}(0, 1)$. Similar to feedback adaptation $\mathbb{E}[\|z_t - z_s\|] = \frac{1}{3}$. For the term $\mathbb{E}[\||t - s\|]$, it is similar to feedforward adaptation, because $s$ is randomly sampled from $[t - L, \cdots, t - 1]$. We have that $\mathbb{E}[\||t - s\|] = \frac{t}{2}$ if $t \leq L$, and $\mathbb{E}[\||t - s\|] = \frac{L+1}{2}$ if $t > L$. We have the expectation for the error bound of random adaptation $B_e^{rnd}$ as shown:

$$\mathbb{E}[B_e^{rnd}] = (K + \hat{K})\mathbb{E}[\|z_t - z_s\|] + \alpha\mathbb{E}[\||t - s\|]$$
$$\leq (K + \hat{K}) \cdot \frac{1}{3} + \alpha \cdot \min(\frac{t}{2}, \frac{L+1}{2}) \tag{39}$$

We mainly consider the final performance $t > L$, then the above equation can be simplified as:

$$\mathbb{E}[B_e^{rnd}] \leq \frac{K + \hat{K}}{3} + \alpha \frac{L + 1}{2} \tag{40}$$

If $L \geq 2$:

$$\frac{K + \hat{K}}{L + 1} + \alpha \frac{L + 1}{2} \leq \frac{K + \hat{K}}{3} + \alpha \frac{L + 1}{2} \tag{41}$$

According to Eq. (37), (40) and (41), we have $\mathbb{E}[B_e^{ff}] \leq \mathbb{E}[B_e^{rnd}]$. Actually if $L = 1$, feedforward adaptation is the same as random adaptation, because the buffer only has one sample. Thus, feedforward adaptation has no greater error bound than random adaptation.

**Expectation of the error bound for full adaptation**. The full adaptation uses all samples from the buffer $[z_{t-L}, \cdots, z_{t-1}]$ to adapt the model. Considering $t > L$, for an arbitrary sample $z_s \sim [z_{t-L}, \cdots, z_{t-1}]$, we have $\mathbb{E}[\|z_t - z_s\|] = \frac{1}{3}$, because all samples in the buffer are independent uniformly-random variables. Considering the final performance $t > L$, we derive the expectation for the error bound of full adaptation $B_e^{fl}$ as shown:

$$\mathbb{E}[B_e^{fl}] = (K + \hat{K})\mathbb{E}[\frac{1}{L}\sum_{s=t-L}^{t-1}\|z_t - z_s\|] + \alpha\mathbb{E}[\frac{1}{L}\sum_{s=t-L}^{t-1}|t - s|]$$
$$= (K + \hat{K}) \cdot \frac{1}{L}\sum_{s=t-L}^{t-1}\mathbb{E}[\|z_t - z_s\|] + \alpha\mathbb{E}[\frac{1}{L}\sum_{i=1}^{L}i]$$
$$= (K + \hat{K}) \cdot \frac{1}{L}\sum_{s=t-L}^{t-1}\mathbb{E}[\|z_t - z_s\|] + \alpha\frac{L+1}{2}$$
$$\leq (K + \hat{K}) \cdot \frac{1}{L}\sum_{s=t-L}^{t-1}\frac{1}{3} + \alpha\frac{L+1}{2}$$
$$\leq (K + \hat{K}) \cdot \frac{1}{3} + \alpha\frac{L+1}{2} \tag{42}$$

As can be seen, the expected error bound of full adaptation is the same as the error bound of random adaptation. It is reasonable because full adaptation usually averages out individual differences between samples. Thus we have a similar conclusion as random adaptation, that feedforward adaptation has no greater error bound than full adaptation. We have a conclusion below.

**Lemma 3**. Given a linear time-variant function (may not be a slowly varying function) Eq. (32), the expected error bound of the feedforward adaptation is smaller than the full adaptation and random adaptation, with the proper buffer size (e.g. $L > 2$).

Therefore, in the following comparison, we only compare the feedforward and feedback adaptation.

**When is feedforward better than feedback**. Similar to the worst-case analysis, we derive the conditions that feedforward adaptation has a smaller error bound than feedback adaptation. To find the condition for $\mathbb{E}[B_e^{ff}] \leq \mathbb{E}[B_e^{fb}]$, we solve the following inequality

$$\sqrt{2\alpha(K + \hat{K})} \leq \frac{K + \hat{K}}{3} + \alpha \quad \text{and} \quad L = \sqrt{\frac{2(K + \hat{K})}{\alpha}} - 1 \geq 1 \tag{43}$$

Then we obtain a $\delta$ time condition on linear time-variant function for feedforward adaptation:

$$\alpha \leq \alpha^\star = \frac{4 - \sqrt{12}}{6}(K + \hat{K}) \approx 0.089(K + \hat{K}) \tag{44}$$

As summary, if time-varying factor $\alpha$ of linear time-variant function Eq. (32) is smaller, specifically $\alpha \leq \frac{4-\sqrt{12}}{6}(K + \hat{K})$, feedforward adaptation has smaller error bound than feedback adaptation. $\alpha^\star = \frac{4-\sqrt{12}}{6}(K + \hat{K})$ is a threshold for selecting whether feedforward or feedback. Thus, we proved the Lemma 1 in section 3.4.

We can extend the above conclusion to a slightly larger set of functions. $\alpha$ can be time-dependent, but the supremum of it needs to be small. Consider the function $f_t(z_t) = g(z_t) + h(t)$, and let $\alpha_t = \frac{\partial f_t}{\partial t} = \frac{\partial h}{\partial t}$. Similar to the above analysis on linear time-variant function Eq. (32), we have a conclusion below.

**Lemma 4**. Given a function $f_t$, let $\alpha_t = \frac{\partial f_t}{\partial t}$. Let $K$ and $\hat{K}$ denote the Lipschitz constant for ground-truth $f_t$ and learnable function (e.g. neural networks) $\hat{f}(\theta)$. If $\sup(\alpha_t) \leq \frac{4-\sqrt{12}}{6}(K + \hat{K})$, with the proper buffer size, feedforward adaptation achieves smaller error bound than feedback adaptation.

Combing Lemma 3 with Lemma 4, we have a conclusion about the effectiveness of the feedforward adaptation method.

**Lemma 5**. Given a function $f_t$, let $\alpha_t = \frac{\partial f_t}{\partial t}$. Let $K$ and $\hat{K}$ denote the Lipschitz constant for ground-truth $f_t$ and learnable function (e.g. neural networks) $\hat{f}(\theta)$.

(a) With a buffer size $L > 2$, feedforward adaptation always has a smaller error bound than full adaptation and random adaptation.

(b) If $\sup(\alpha_t) \leq \frac{4-\sqrt{12}}{6}(K + \hat{K})$, with the proper buffer size $L = \sqrt{\frac{2(K+\hat{K})}{\alpha}} - 1$, feedforward adaptation achieves a smaller error bound than feedback adaptation.

### C.2.2 Prediction Error of online adaptation on Time invariant polynomial

In this section, we compare the error (not error bound) of feedforward and feedback adaptation methods on time-invariant polynomial systems (functions). Consider a time-invariant polynomial ground-truth function $f_t(z_t)$ with the input of random variable sampled from the uniform distribution $\mathcal{U}(0, 1)$.

$$z_t \sim \mathcal{U}(0, 1) \tag{45}$$

$$y_{t+1} = f_t(z_t) = \sum_{i=1}^{N} W_i z_t^i \tag{46}$$

Assume our parameterized prediction model is a linear projection:

$$\hat{y}_t = \hat{f}(V_t, z_t) = V_t z_t \tag{47}$$

In online adaptation, we use the critical pair $(z_s, y_{s+1})$ to optimize the linear prediction model at time step $t$. Then we have a parameter of

$$V_t = \arg\min_V \|V z_s - y_{s+1}\| = \frac{y_{s+1}}{z_s} = \frac{\sum_{i=1}^{N} W_i z_s^i}{z_s} = \sum_{i=0}^{N-1} W_{i+1} z_s^i \tag{48}$$

Then the absolute prediction error ($l_1$ norm) of online adaptation is shown below:

$$
\begin{aligned}
e_{t+1} = |y_{t+1} - \hat{f}(V_t, z_t)| = |y_{t+1} - V_t z_t| = |\sum_{i=1}^{N} W_i z_t^i - \sum_{i=0}^{N-1} W_{i+1} z_s^i \cdot z_t| \\
= |z_t| \cdot |\sum_{i=0}^{N-1} W_{i+1}(z_t^i - z_s^i)| = |z_t| \cdot |\sum_{i=0}^{N-1} W_{i+1}(\sum_{j=0}^{i-1} z_t^{i-j-1} z_s^j)(z_t - z_s)| \\
= |z_t| \cdot |z_t - z_s| \cdot |\sum_{i=0}^{N-1} W_{i+1} \sum_{j=0}^{i-1} z_t^{i-j-1} z_s^j|
\end{aligned}
\tag{49}
$$

We consider the expectation of the prediction error. We have the following property in statistics:

$$
\mathbb{E}(z_t^n) = \frac{1}{n+1} \text{ for } z_t \sim \mathcal{U}(0,1)
\tag{50}
$$

For simplicity, assume $W_i > 0$. The expectation of the prediction error is shown below.

$$
\mathbb{E}(e_{t+1}) = \mathbb{E}(z_t) \cdot \mathbb{E}(|z_t - z_s|) \cdot \sum_{i=0}^{N-1} W_{i+1} \sum_{j=0}^{i-1} \mathbb{E}[z_t^{i-j-1} z_s^j]
\tag{51}
$$

Then we use Eq. (51) to derive the expectation of the error for feedback and feedforward adaptation methods. The core is measurement of $\mathbb{E}[\|z_t - z_s\|]$ and $\mathbb{E}[z_t^{i-j-1} z_s^j]$.

**Expectation of the prediction error for feedback adaptation**. In feedback adaptation, $z_s = z_{t-1}$ the current sample $z_t$ and the selected sample $z_s$ are independent random variables from $\mathcal{U}(0,1)$. We have $\mathbb{E}(|z_t - z_s|) = \frac{1}{3}$ (Eq. (35)). Then we derive an expectation of the prediction error for feedback adaptation:

$$
\begin{aligned}
\mathbb{E}(e_{t+1}^{fb}) &= \frac{1}{2} \cdot \frac{1}{3} \sum_{i=0}^{N-1} W_{i+1} \sum_{j=0}^{i-1} \mathbb{E}[z_t^{i-j-1} x_s^j] = \frac{1}{6} \sum_{i=0}^{N-1} W_{i+1} \sum_{j=0}^{i-1} \mathbb{E}[z_t^{i-j-1}] \cdot \mathbb{E}[z_s^j] \\
&= \frac{1}{6} \sum_{i=0}^{N-1} W_{i+1} \sum_{j=0}^{i-1} \frac{1}{i-j} \frac{1}{j+1} \\
&\geq \frac{1}{6} \sum_{i=0}^{N-1} W_{i+1} \sum_{j=0}^{i-1} \frac{2}{(i+1)^2} = \frac{1}{3} \sum_{i=0}^{N-1} W_{i+1} \frac{i}{(i+1)^2} \\
&\geq \frac{1}{3} \frac{N-1}{N^2} \sum_{i=1}^{N-1} W_{i+1}
\end{aligned}
\tag{52}
$$

As can be seen, the prediction error of feedback adaptation cannot converge. The expected prediction error $\mathbb{E}(e_{t+1}^{fb})$ in each step is lower bounded by a non-zero constant.

**Expectation of the prediction error for feedforward adaptation**. In feedforward adaptation, the selected samples $z_s$ are the most similar samples to the current observation $z_t$. We have $\mathbb{E}(|z_t - z_s|) \leq \max(\frac{1}{t}, \frac{1}{L+1})$ (Eq. (37)). Then we derive an expectation of the prediction error

for feedforward adaptation:

$$
\mathbb{E}(e_{t+1}^{ff}) \leq \frac{1}{2} \max(\frac{1}{t}, \frac{1}{L+1}) \sum_{i=0}^{N-1} W_{i+1} \sum_{j=0}^{i-1} \mathbb{E}[z_t^{i-j-1} z_s^j]
$$

$$
\leq \frac{1}{2} \max(\frac{1}{t}, \frac{1}{L+1}) \sum_{i=0}^{N-1} W_{i+1} \sum_{j=0}^{i-1} \mathbb{E}[z_t^{i-j-1} z_t^j]
$$

$$
\leq \frac{1}{2} \max(\frac{1}{t}, \frac{1}{L+1}) \sum_{i=0}^{N-1} W_{i+1} \sum_{j=0}^{i-1} \mathbb{E}[z_t^{i-1}]
$$

$$
\leq \frac{1}{2} \max(\frac{1}{t}, \frac{1}{L+1}) \sum_{i=0}^{N-1} W_{i+1} \sum_{j=0}^{i-1} \frac{1}{i}
$$

$$
\leq \frac{1}{2} \max(\frac{1}{t}, \frac{1}{L+1}) \sum_{i=1}^{N-1} W_{i+1} \tag{53}
$$

When we consider the final performance,

$$
\lim_{L\to\infty, t\to\infty} \mathbb{E}(e_{t+1}^{ff}) \leq \lim_{L\to\infty, t\to\infty} \frac{1}{2} \max(\frac{1}{t}, \frac{1}{L+1}) \sum_{i=1}^{N-1} W_{i+1} = 0 \tag{54}
$$

As can be seen, the expectation of the prediction error of feedforward compensation converges to zero with a large buffer size $L$. Thus, we proved the Lemma 2 in section 3.4.

**Expectation of the prediction error for full (or random) adaptation**. For full adaptation or random adaptation, we have $\mathbb{E}(|z_t - z_s|) = \frac{1}{3}$ (Eq. (42)), which is the same as feedback adaptation. Then we can see that the expectation of the prediction error for full (or random) adaptation is the same as feedback adaptation:

$$
\mathbb{E}(e_{t+1}^{fl}) = \frac{1}{2} \cdot \frac{1}{3} \sum_{i=0}^{N-1} W_{i+1} \sum_{j=0}^{i-1} \mathbb{E}[z_t^{i-j-1} x_s^j]
$$

$$
\geq \frac{1}{3} \frac{N-1}{N^2} \sum_{i=1}^{N-1} W_{i+1} \tag{55}
$$

As can be seen, the prediction error of full adaptation cannot converge. Thus, feedforward adaptation is provably better than full adaptation in the time-invariant polynomial system.

**When is feedforward better than feedback**. In the previous appendix C.2.1, we already know that feedforward adaptation has a smaller error bound than feedback adaptation on approximate time-invariant system ($\alpha \approx 0$). In this section, we further prove that for the time-invariant polynomial system, feedforward adaptation has a smaller expected prediction error (not an error bound) than feedback adaptation, full adaptation, and random adaptation methods. As the result, feedforward adaptation is better than feedback adaptation (or full adaptation, random adaptation) for time-invariant or slowly time-variant systems.

## D  ADDITIONAL EXPERIMENTS

### D.1  SYNTHETIC EXPERIMENTS: LINEAR TIME-VARYING FUNCTION

To evaluate Lemma 1 in section 3.4, we consider the following time-varying function.

$$
z_t \sim \mathcal{U}(0, 1) \tag{56}
$$
$$
y_{t+1} = f_t(z_t) = \sin z_t + \alpha t \tag{57}
$$

Our parameterized prediction model is a one-layer perception with Sigmoid activation function.

$$
\hat{y}_t = \hat{f}(V_t, b_t; z_t) = S(V_t z_t) + b_t = \frac{1}{1 + e^{-V_t z_t}} + b_t \tag{58}
$$

We have The Lipschitz constant $K$ and $\hat{K}$ for the ground-truth function $f_t$ and the one-layer perception $\hat{f}$:

$$K = \sup |\frac{\partial f_t}{\partial z_t}| = \sup |\cos(z_t)| = 1 \tag{59}$$

$$\hat{K} = \sup(|\frac{\partial \hat{f}}{\partial z_t}|) = \sup |V_t \cdot S(V_t z_t) \cdot (1 - S(V_t z_t))| = 0.25 \sup |V_t| \tag{60}$$

We use SGD as an optimizer in feedback and feedforward adaptation. During training, we keep the $\|V_t\|$ bounded, i.e. $\|V_t\| \leq 1$, then $\hat{K} = \frac{1}{4}$. We use Lemma 1 to calculate the error bound for feedback and feedforward adaptation:

$$\mathbb{E}[B_e^{fb}] = \frac{5}{12} + \alpha \tag{61}$$

$$\mathbb{E}[B_e^{ff}] \leq \frac{5}{4(L+1)} + \alpha \frac{L+1}{2} \tag{62}$$

In feedforward adaptation, when $L = \sqrt{\frac{5}{2\alpha}} - 1$, we get the minimum error bound:

$$\mathbb{E}[B_e^{ff}] \leq \sqrt{\frac{5}{2}\alpha} \ \text{ with } \ L = \sqrt{\frac{5}{2\alpha}} - 1 \tag{63}$$

Then we calculate the threshold $\alpha^\star$. If $\alpha \leq \alpha^\star$, feedforward adaptation has a smaller error bound.

$$\alpha^\star = \frac{4 - \sqrt{12}}{6}(K + \hat{K}) \approx 0.089(K + \hat{K}) \approx 0.11 \tag{64}$$

$$L^\star = \sqrt{\frac{5}{2\alpha^\star}} - 1 \approx 3 \tag{65}$$

The experimental evaluation of the above theoretical results is shown in section 4.1.

### D.2 SYNTHETIC EXPERIMENTS: TIME-INVARIANT QUADRATIC FUNCTION

To evaluate Lemma 2 in section 3.4, we consider the following quadratic function with random input.

$$z_t \sim \mathcal{U}(0, 1) \tag{66}$$

$$y_{t+1} = f_t(z_t) = z_t^2 \tag{67}$$

Our parameterized prediction model is a linear projection:

$$\hat{y}_t = \hat{f}(V_t, z_t) = V_t z_t \tag{68}$$

Set the buffer size $L = 99$. According to Lemma 2, the expected prediction error for feedback and feedforward adaptation are:

$$\mathbb{E}[e_{t+1}^{fb}] \geq \frac{1}{3}\frac{N-1}{N^2}\sum_{i=1}^{N-1} W_{i+1} = \frac{1}{12} \tag{69}$$

$$\mathbb{E}(e_{t+1}^{ff}) \leq \frac{1}{2}\max(\frac{1}{t}, \frac{1}{L+1})\sum_{i=1}^{N-1} W_{i+1} = \frac{1}{2}\max(\frac{1}{t}, \frac{1}{L+1}) = \max(\frac{1}{2t}, \frac{1}{200}) \tag{70}$$

Fig. 3a shows the prediction error for feedback and feedforward adaptation. As can be seen, the error curve for feedback adaptation is not converged. While the error curve for feedforward adaptation converges by the trend of $\frac{1}{2t}$. The experimental results are consistent with Eq. (69) and (70). Fig. 3a shows the prediction results for feedforward adaptation. The dashed blue region denotes the estimated uncertainty in section 3.3. In the figure, the ground truth is always within the estimated uncertainty, which validifies the proposed uncertainty estimation.

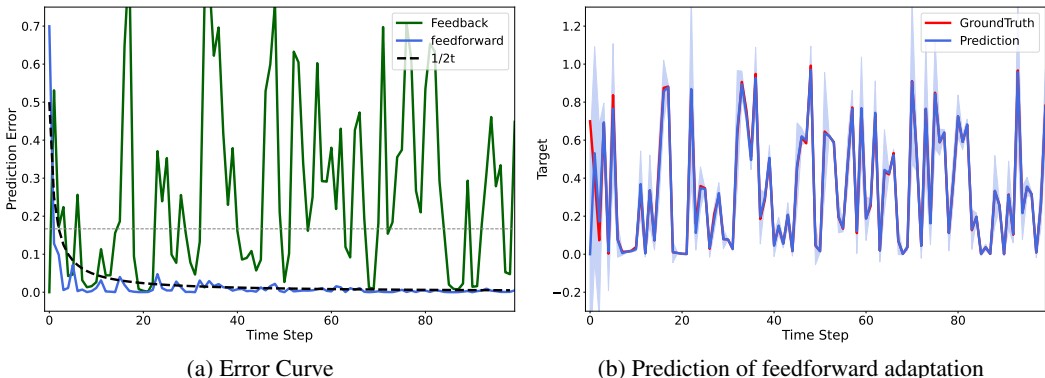

(a) Error Curve

(b) Prediction of feedforward adaptation

Figure 3: Error curve and prediction results on quadratic function.

### D.3 EXPERIMENTAL DESIGN FOR REAL-WORLD EXPERIMENTS

**Dataset**.

- **ETT** Zhou et al. (2021) dataset contains the data collected from electricity transformers, including load and oil temperature that are recorded every 15 minutes between July 2016 and July 2018. Which consists of two hourly-level datasets (ETTh) and two 15-minute-level datasets (ETTm). In our experiments, we used the first hourly-level dataset ETTh1 as a univariance prediction task.
- **Exchange-Rate** Lai et al. (2018) records the daily exchange rates of eight countries from 1990 to 2016.
- **ILI** [2] describes the ratio of patients seen with ILI and the total number of patients. Which includes the weekly recorded influenza-like illness (ILI) patients data from the Centers for Disease Control and Prevention of the United States between 2002 and 2021.
- **THOR** Rudenko et al. (2020) is a public dataset of human motion trajectories, recorded in a controlled indoor experiment. Which includes the motion trajectories with diverse and accurate social human motion data in a shared indoor environment. In our experiments, we use No. $2 \sim 6$ agent's trajectory as a train set, No. $7 \sim 8$ as a validation set, and No. $9 \sim 10$ agent's trajectory as a test set.

**Backbone models**. We include two models as our learnable prediction function $\hat{f}(\theta, :)$: Informer Zhou et al. (2021) and simple MLP (multi-layer perception).

- **Informer** Zhou et al. (2021) is a widely used transformer-based time-series prediction model. Which extends the Transformer with KL-divergence based ProbSparse attention.
- **MLP** is a simple but robust baseline for time-series prediction. Our MLP consists of 2 layers. The first layer can be considered as Encoder $Z_t = W \cdot X_t$. After the encoder, MLP has a layer normalization, activation function and a final linear projection $Y_{t+1} = V \cdot \text{Relu}(\text{LayerNorm}(Z_t))$. The layer normalization and the final projection can be served as a decoder. Note that, we did not flatten the input for MLP, the expression $Z_t = W \cdot X_t$ is a linear layer along the temporal axis.

**Hyperparameters**. For offline training, we follow the strategy in Zeng et al. (2022). In adaptation, we set the learning rate of SGD as $\eta = 0.1$ and set the EKF hyperparameters as $p_0 = 0.1, \lambda = 1$. Buffer size for feedforward adaptation is $L = 1000$. For ETTh1 dataset, we use $I = 192$ step recent observations to predict future $O = 192$ step output data. For the Exchange-Rate dataset, we use $I = 96$ step recent observations to predict future $O = 192$ step output data. For the ILI dataset, we use $I = 36$ step recent observations to predict future $O = 36$ step output data. For the THOR dataset, we use $I = 20$ step recent observations to predict future $O = 20$ step output data. For uncertainty estimation, we set $\tilde{\delta} = 0, \tilde{K} = 1$.

### D.4 EXPERIMENTAL RESULTS ON EKF OPTIMIZERS

We conduct experiments on SGD and EKF as optimizers in adaptation. The descriptions of SGD and EKF are shown in appendix B. In this section, we report the detailed results of the experiment on the

---

[2]https://gis.cdc.gov/grasp/fluview/fluportaldashboard.html

EKF optimizer, as shown in table 2. Similar to results on the SGD on table 1, feedforward adaptation still achieves the best results on all four datasets with EKF optimizer. Specifically, feedforward adaptation outperforms feedback adaptation by $5.9\%, 15.1\%, 24.1\%$, and $2.6\%$ in terms of average MSE on four datasets respectively. The performance gain on the ILI dataset is the most obvious. It is reasonable because the incidence rate of common diseases in a region is a slow time-varying system, the future value is mostly dependent on the current input.

Table 2: Performance comparison between the proposed feedforward adaptation method and other baselines with EKF optimization. Avg denotes the average results of two models (MLP and Informer). The last row denotes the performance gain of feedforward adaptation over feedback adaptation.

| Method \ Dataset | | ETTh1 | | Exchange | | ILI | | THOR | |
|---|---|---|---|---|---|---|---|---|---|
| Adaptation | Model | MSE | MAE | MSE | MAE | MSE | MAE | MSE | MAE |
| w/o Adaptation | MLP | 0.195 | 0.371 | 0.549 | 0.540 | 4.348 | 1.413 | 0.135 | 0.117 |
| | Informer | 0.211 | 0.389 | 1.128 | 0.858 | 4.942 | 1.531 | 0.137 | 0.171 |
| | Avg | 0.203 | 0.380 | 0.839 | 0.699 | 4.645 | 1.472 | 0.136 | 0.144 |
| Random Adaptation | MLP | 0.174 | 0.335 | 0.404 | 0.468 | 3.318 | 1.262 | 0.123 | 0.149 |
| | Informer | 0.191 | 0.352 | 0.847 | 0.684 | 3.573 | 1.328 | 0.277 | 0.313 |
| | Avg | 0.182 | 0.343 | 0.625 | 0.576 | 3.445 | 1.295 | 0.200 | 0.231 |
| Feedback Adaptation | MLP | 0.129 | 0.288 | 0.356 | 0.459 | 3.873 | 1.341 | 0.107 | 0.106 |
| | Informer | 0.109 | 0.249 | 0.359 | 0.462 | 4.320 | 1.401 | 0.124 | 0.168 |
| | Avg | 0.119 | 0.268 | 0.357 | 0.460 | 4.110 | 1.371 | 0.116 | 0.137 |
| Feed Forward Adaptation | MLP | 0.123 | 0.280 | 0.352 | 0.450 | 2.847 | 1.154 | 0.102 | 0.103 |
| | Informer | 0.102 | 0.248 | 0.254 | 0.327 | 3.395 | 1.244 | 0.124 | 0.167 |
| | Avg | 0.112 | 0.264 | 0.303 | 0.388 | 3.121 | 1.199 | 0.113 | 0.135 |
| Gain of Feedforward Over Feedback | | 5.9% | 1.5% | 15.1% | 15.6% | 24.1% | 12.5% | 2.6% | 1.5% |

### D.5 STUDY OF THE SAMPLE SELECTION STRATEGY OF DIFFERENT ADAPTATION METHODS

We will show that the sample selection strategy in the proposed feedforward adaptation method could intrinsically extract the periodicity of the input data. The experiments on the ETTh1 dataset were shown in Fig. 2. In this section, we show the results of ILI, Exchange, THOR datasets. In our experiment, we set the buffer size $L = 1000$. We use SGD optimizer to adapt the MLP model with different adaptation methods.

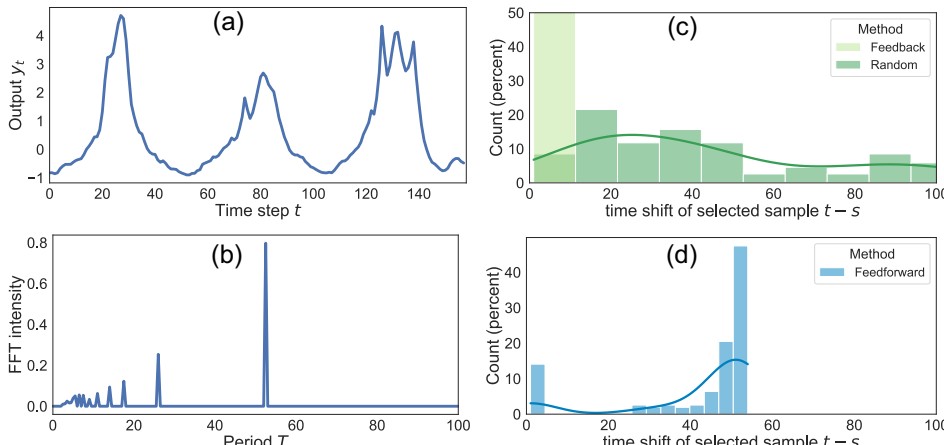

Figure 4: Experiments on ILI data. (a) Time series output; (b) FFT period analysis; (c) Time shift $t-s$ between current sample $Z_t$ and selected sample $Z_s$ in feedback and random adaptation; (d) Time shift $t - s$ in feedforward adaptation; (e) Online prediction error. (f) Sample difference $\|Z_t - Z_s\|$ between current sample $Z_t$ and adapted sample $Z_s$.

**ILI dataset**. Figure 4(a) shows the ILI time-series data (1st dimension of the output), and Fig. 4(b) is the FFT (Fast Fourier Transformation) period analysis of the ILI data. As can be seen ETTh1 has roughly $T \approx 50$ repetition periods. Random adaptation randomly selects samples from the

buffer, then the time shift between current time-step $t$ and the time-step for selected samples $s$ is $(t - s) \sim [1, L]$; Feedback adaptation only selects the last time sample to optimization, then $(t - s) = 1$. This can be found from the distribution of time shift $(t - s)$ in Fig. 4(c). For the proposed feedforward adaptation in Fig. 4(d), many samples were selected from $(t - s) \approx 50$ steps earlier, which corresponds to the repetition period of $T \approx 50$. Because feedforward adaptation selects the most similar samples to the current sample, which helps to extract the hidden periodicity of the input signal over time. Thus the distribution of $t - s$ is similar to FFT period analysis.

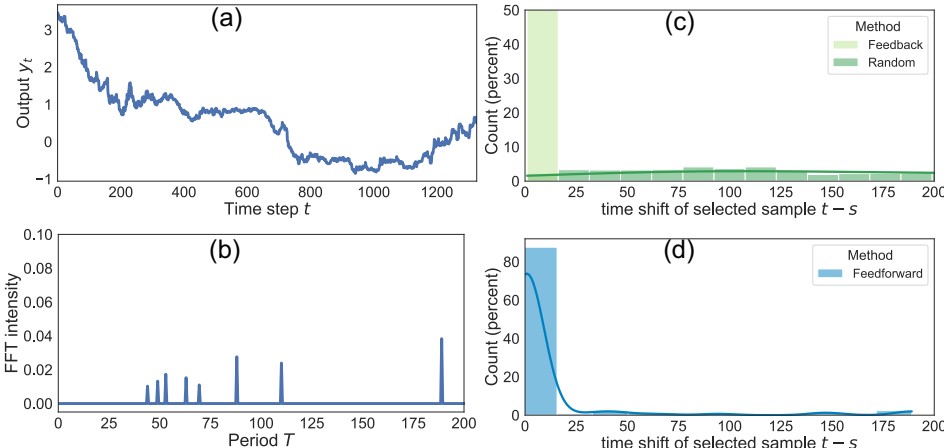

Figure 5: Experiments on Exchange data. (a) Time series output; (b) FFT period analysis; (c) Time shift $t - s$ between current sample $Z_t$ and selected sample $Z_s$ in feedback and random adaptation; (d) Time shift $t - s$ in feedforward adaptation; (e) Online prediction error. (f) Sample difference $\|Z_t - Z_s\|$ between current sample $Z_t$ and adapted sample $Z_s$.

**Exchange dataset**. Figure 5(a) shows the Exchange time-series data (6th dimension of the output), and Fig. 5(b) is the FFT (Fast Fourier Transformation) period analysis of the Exchange data. As can be seen Exchange has no apparent periodicity. (The intensity of the FFT signal is too low, for example, the intensity of at the $T \approx 90$ is about 0.02). In this case, Feedforward adaptation in Fig. 5(d), is likely to select the recent samples like feedback adaptation in Fig. 5(c). Because the most recent sample is a similar sample itself. Even for this kind of no-periodic system, feedforward adaptation still outperforms feedback adaptation, because it has more freedom to select samples. At least feedforward adaptation can select samples exactly the same as feedback adaptation.

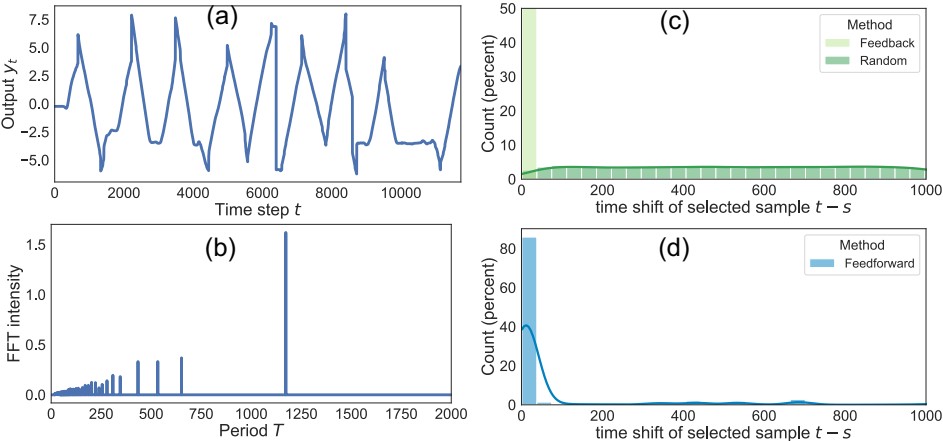

Figure 6: Experiments on THOR data. (a) Time series output; (b) FFT period analysis; (c) Time shift $t - s$ between current sample $Z_t$ and selected sample $Z_s$ in feedback and random adaptation; (d) Time shift $t - s$ in feedforward adaptation; (e) Online prediction error. (f) Sample difference $\|Z_t - Z_s\|$ between current sample $Z_t$ and adapted sample $Z_s$.

**THOR dataset**. Figure 6(a) shows the THOR time-series data (1st dimension of the output), and Fig. 6(b) is the FFT (Fast Fourier Transformation) period analysis of the THOR data. As can be seen

THOR has roughly $T \approx 1200$ repetition periods. But the buffer size is $L = 1000 < T$. In this case, we only store the recent $L = 1000$ samples in the buffer, but the period is more larger $T \approx 1200 > L$, so the feedforward adaptation cannot extract the periodicity. In this case, Feedforward adaptation Fig. 6(d), is likely to select the recent samples like feedback adaptation in Fig. 6(c). Because the most recent sample is a similar sample itself within the buffer. In the future, we will investigate the more efficient sampling and buffer storing strategy to extract the very long time-dependency and periods.

### D.6 PREDICTION OUTPUT AND UNCERTAINTY

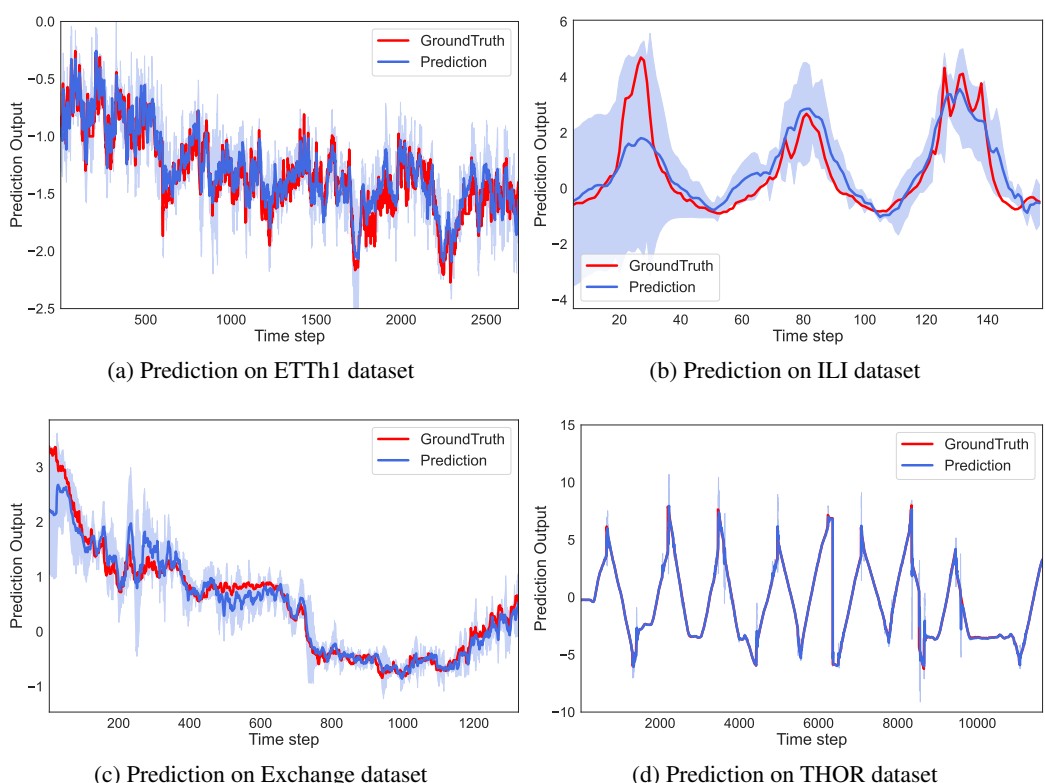

(a) Prediction on ETTh1 dataset

(b) Prediction on ILI dataset

(c) Prediction on Exchange dataset

(d) Prediction on THOR dataset

Figure 7: Prediction output and uncertainty estimation of the proposed feedforward adaptation on (a) ETTh1 dataset, (b) ILI dataset (1st dimension of the output), (c) Exchange dataset (6th dimension of the output), and (d) THOR dataset (1st dimension of the output). The blue dashed region represents the estimated uncertainty of the prediction.

One of the advantages of the proposed feedforward adaptation is it could provide uncertainty estimation as shown in algorithm 2. Figure 7 shows the prediction output (blue curve), ground truth label (red curve), and uncertainty estimation (blue dashed region) on four different datasets. As can be seen, in most cases, the estimated uncertainty covers the real ground truth value. Which validates the effectiveness of the proposed uncertainty estimation.

