# OpenReview forum: "Online Continual Learning with Feedforward Adaptation"
_ICLR.cc/2023/Conference — Submitted to ICLR 2023_

### Official Review · Reviewer_Qtjr · 2022-10-15

**Confidence:** 4
**Correctness:** 4
**Technical Novelty And Significance:** 2
**Empirical Novelty And Significance:** 2
**Recommendation:** 6

**Clarity, Quality, Novelty And Reproducibility:**

The proposed approach is clearly explained and it seems novel. The quality of the study needs improvement.

**Strength And Weaknesses:**

The studied topic is interesting and it indeed seems to be the most viable learning scenario in practice in many applications.
The contributions are clearly mentioned and the introduction motivates the reader to the topic.
The proposed approach is interesting and specially introducing an approach uncertainty quantification can be very useful.

However, when we get to the experimental results, the experiments lack the claims made in the paper.
I would expect to see some experiments related to the uncertainty measurement for the real datasets.
Also, for the real datasets, there is no experiment to study for each dataset, how much on average (or one single run), the critical selected examples, differ from the latest examples, and illustrate how this deviation relates to gaining in feed-forward in compare with feedback.  Also you could check for each dataset, how many times the critical examples, where actually the latest ones, or analyze the time invariance element empirically.

In Page 3, “Let f_t  denotes the ground-truth predictor … that generates the ground-truth input-output tuples ”, please make it clear that it is only predicting the output not the tuple (not a generative approach).

In table 1, you could use bold face and underlined for the best and second best results, to make it easier to read.

There are typos and ambiguous sentences in the manuscript, such as:
 - Page 3: “Online adaptation continually learning from streaming data, which is belong…”  wrong sentence, please rewrite it.
- Page 4, the last sentence of the page is incomplete. “check the details in.”
- Page 5, eq(5) $\hat {K}$, is not defined.
- Several times in the main manuscript you referred to an equation that was in in appendix, whereas you had that same equation nearby too. Examples: Page 5, your referred to Eq. 29, instead of Eq. 8.
Page 6, Eq. 31 instead of Eq. 15. Page 7, Eq. 60, instead of Eq. 19 and Eq. 61 instead of Eq. 20.
- In algorithm 2, line4, it should be s* in the subscript not s.
- In Lemma 2, B and $\hat {B}$ should be K and $\hat {K}$.



**Summary Of The Paper:**

In this study an online feed-forward adaptation approach is proposed to address distribution shift in  upcoming test data with respect to the training data. The core idea in feed-forward adaptation is finding a critical example from the memory, that is the most related to the current example, and adjust the model to predict the desired output of that critical example (in the past). This approach has been compared with the feedback adaptation approach in which the prediction model is updated according to the loss measured for the latest received ground truth. The authors also provided an analysis of the uncertainty of the prediction.


**Summary Of The Review:**

In my opinion, this manuscript is not ready yet to be published at ICLR. However, it depends on how the authors address the raised issues.

---

> ### Author Response · Authors · 2022-11-10
> **Response to reviewer Qtjr**
>
> We would like to greatly thank the reviewer for the constructive suggestions.  Please find the detailed responses below.
>
> *Q1.  I would expect to see some experiments related to uncertainty measurement for real datasets.*
>
> A1. Thank you for the suggestions. We conduct experiments of prediction and uncertainty estimation in appendix D.6. In the experiments, the ground truth is located within the estimated uncertainty, which validifies the proposed uncertainty estimation.
>
> *Q2.  There is no experiment to study the selected examples.*
>
> A2. Thanks for the constructive suggestion.  In section 4.2 and appendix D.5 of the revised version, we conduct a study about the sample selection strategy of different adaptation methods. The study shows that the sample selection strategy in the feedforward adaptation method could intrinsically extract the periodicity of the input data, while other methods cannot.
>
> *Q3. There are typos and ambiguous sentences in the manuscript.*
>
> A3. We fixed these issues in the revised manuscript.
>
> Please let me know if you need further details.

---

> > ### Comment · Reviewer_Qtjr · 2022-11-25
> > **Esperience replay**
> >
> > Thank you very much for addressing the previous comments.
> > I have a question regarding the relation of your setup and approach with experience replay approaches such as "https://arxiv.org/pdf/1902.10486.pdf" and specially "https://arxiv.org/pdf/1908.04742.pdf" that it uses the samples that cause the most forgetting instead of random sampling in the past. Your approach seems very similar to their and at least you could include a comparison, or mentioning those approaches and explain how your approach differentiates from their. Thank you again for your reposes.

---

> > > ### Author Response · Authors · 2022-11-25
> > > **Response to the experience replay**
> > >
> > > Thank you for your suggestions.
> > > Experience replay is a typical case of the Rehearsal-based approach, and the rehearsal methods were mentioned in related works of our paper.  We will add a more detailed comparison of Tiny-Mem-ER [1] and MIR [2] in the revision. The Tiny-Mem-ER,  MIR, and the proposed feedforward adaptation can be considered as the experience replay methods. The main differences are that Tiny-Mem-ER and MIR focus on classification tasks and batch continual learning, while our method focuses on regression (time series) and online sample-level continual learning. The detailed discussion and comparison are shown below.
> > > 1) In Tiny-Mem-ER [1], Reservoir Sampling is actually the same as our random adaptation method. Ring Buffer Sampling implements FIFO (first-input-first-output) buffer within each class. But our tasks do not have classes, because we focus on time series prediction tasks. If we assume all time-series data belongs to one class, then Ring FIFO Buffer is very similar to feedback adaptation.  K-Means Sampling and Mean of Features (MoF) Sampling rely on the centroids or average feature vector for each class. Because our tasks are regression tasks (no class here), the  K-Means Sampling and MoF sampling are not applicable.
> > > 2) MIR is still focused on image classification tasks and batch continual learning. More importantly,  MIR pays more attention to generalization ability, so it retrieves samples that suffer from an increase in loss at the current training step. Because we cannot access labels in real-world image classification applications. However, in online time-series prediction, we focus on the ability to rapidly learn and adapt in the presence of non-stationarity instead of generalization ability [3]. One of the reasons is that in time-series prediction, the input signal $x_t$ at time-step $t$ is a partial label of the previous signal $x_{t-1}$. That means, in real-world time-series prediction, we can access partial labels with a time lag.  In order to rapidly learn the current data and perform better in the future few steps (instead of globally better like MIR), the proposed feedforward adaptation selects the most similar samples from the buffer.   Then the proposed feedforward adaptation performs better in the local region of current samples not globally.
> > > 3) Tiny-Mem-ER and MIR do not have any theoretical guarantee. In our work, the proposed feedforward adaptation has provably converged to zero prediction error in a time-invariant polynomial system (Lemma 2), and its prediction error is bounded in a slowly time-variant system (Lemma 1).
> > >
> > > We will add the above discussion in the revision.
> > >
> > >
> > > [1] Arslan Chaudhry, et al., "On tiny episodic memories in continual learning",arxiv 2019.
> > > [2] Rahaf Aljundi, et al., "Online continual learning with maximal interfered retrieval", NeurIPS 2019.
> > > [3] Yujiao Cheng, et al. "Human motion prediction using semi-adaptable neural networks." American Control Conference (ACC), 2019.

---

> > > > ### Comment · Reviewer_Qtjr · 2022-11-28
> > > > **existing time series experience replay study**
> > > >
> > > > Thank you very much for your response.
> > > > If you take a look in this manuscript "https://arxiv.org/abs/2202.11672", actually they have the ER as the baseline to compare with their presented method and it is actually for time series forecasting and also in an online continual learning setup.
> > > > Apart from that, I think the uncertainty measurement that you have provided in your approach in valuable. However, you did not mention other approaches that can be used to evaluate the uncertainty and how your approach is differentiated from the others for confidence interval measurement and actually even compare  them or somehow justify the proposed strategy.

---

> > > > > ### Author Response · Authors · 2022-11-28
> > > > > **Response to the  ER and uncertainty estimation**
> > > > >
> > > > > We would like to greatly thank the reviewer for the constructive suggestions.
> > > > >
> > > > > 1.  Our random adaptation baseline is the same as the  Experience Replay in the paper [1].
> > > > > In section D.1.1 of the FSNet paper [1], the author said "ER deploy a reservoir sampling buffer of 500 samples".  As described in the last response, Reservoir (random) Sampling with a buffer is the same as our random adaptation method. We will add to the revision that, our random adaptation baseline is the same as the  Experience Replay with Reservoir Sampling.
> > > > >
> > > > > 2. FSNet is the feedback adaptation approach with a special network and parameter update strategy.
> > > > > Our paper focuses on the data compensation (or selection) strategy and proposes the feedforward compensation for online continual learning. Which is agnostic to network design.  FSNet [1] proposes an efficient network architecture and per-layer adaptation strategy. Our work is compatible with FSNet. As can be seen from Algorithm 1 of the paper [1], FSNet uses the feedback compensation method, since it uses the latest look-back window $x_t$ to optimize the model.  We can furtherly replace the latest  $x_t$ to the critical sample $x_s^\star$ of feedforward adaptation.  We will add the related work to our revision.
> > > > >
> > > > > 3. The proposed uncertainty estimation is more efficient than many widely used methods.
> > > > > For general network design and time-series prediction tasks, there are two widely used uncertainty estimation methods: Bayesian Methods and Ensemble Methods [2]. In bayesian methods, the model parameters are modeled as random variables. In ensemble methods, the predictions of several models are combined into one prediction. Compared to these widely used methods, the proposed uncertainty estimation has several differences:
> > > > > 3.1) Low computational cost. Bayesian Methods need to approximate Bayesian inference and compute the posterior probabilities. Which have been proven to be difficult as the size of the data and the number of parameters are too large. Ensemble methods train and evaluate several models, which could be memory and computationally inefficient with the increase of parameters. However, the proposed uncertainty estimation could measure uncertainty with a single pass and a single model.
> > > > > 3.2) More special usage. The proposed uncertainty estimation is only used for online adaptation for time-series prediction. But Bayesian Methods and  Ensemble methods can be used for more general cases.
> > > > > 3.3) Theoretically reasonable. As described in the paper, the proposed uncertainty estimation was derived from error bounds. So it has the theoretical guarantee to describe the error bound. However,  applicable Bayesian Methods and  Ensemble methods are not such guarantees, because of the approximation of Bayesian inference and the small number of model ensembles.
> > > > > We will add the above comparison in the revision.
> > > > >
> > > > > [1] Quang Pham, et al. "Learning Fast and Slow for Online Time Series Forecasting." arXiv preprint, 2022.
> > > > > [2] Jakob Gawlikowski, et al. "A survey of uncertainty in deep neural networks." arXiv preprint, 2021.

---

> > > > > > ### Comment · Reviewer_Qtjr · 2022-11-29
> > > > > > **feedforward vs. MIR**
> > > > > >
> > > > > > Thank you very much for your response.
> > > > > > My point here is that in the FSNet paper, for the baseline they have used MIR in an online continual scenario for time series, which matches your setup. As your approach seems very similar to MIR, in my opinion, you need to do a comparison with MIR (if possible) or explain how your approach differentiates from MIR.  Regarding uncertainty, it will be nice to do some comparison there and discuss how your approach for uncertainty identification is superior than the other methods. Anyway, I am willing to increase my vote for this study as the authors already address many issues.
> > > > > > Thank you very much for your attention.

---

> > > > > > > ### Author Response · Authors · 2022-11-29
> > > > > > > **Thank you for the further discussion**
> > > > > > >
> > > > > > > Thank you for the further review.  We will add some detailed comparisons with MIR, and a discussion about the uncertainty will be added.
> > > > > > >
> > > > > > > 1. MIR focus on generalization ability, while our method leans to perform better in the local region.
> > > > > > > MIR focus on classical batch-level continual learning, we consider online sample-level real-time continual learning. MIR pays more attention to generalization ability, so it retrieves samples that suffer from an increase in loss at the current training step. Which is helpful to decrease the loss globally. But in our work, we consider decreasing the loss in the region of current data.   We focus on the ability to rapidly learn and adapt in the presence of non-stationarity instead of generalization ability.  In order to rapidly learn the current data and perform better in the future few steps (instead of globally better like MIR), the proposed feedforward adaptation selects the most similar samples from the buffer.   We will add the comparison.
> > > > > > >
> > > > > > > 2. Discussion about the uncertainty estimation that was shown in the last response will be added to the revision. In short, the advantages of the proposed uncertainty estimation are computationally efficient and theoretically reasonable.
> > > > > > >
> > > > > > > Thanks for the advice and time again!

---

### Official Review · Reviewer_kCCH · 2022-10-28

**Confidence:** 4
**Correctness:** 4
**Technical Novelty And Significance:** 2
**Empirical Novelty And Significance:** 2
**Recommendation:** 5

**Clarity, Quality, Novelty And Reproducibility:**

Overall, the paper is well-written and also has some novelties although it is limited.

**Strength And Weaknesses:**

Strength: The idea of feedforward adaptation is interesting in online continual learning.

Weaknesses:

- The paper does not motivate the benefit of selecting the most similar data samples well. One simple solution is the learner adapt the model using all data samples in the buffer. Specifically, the paper does not consider the problem of online decision making and the learner only updates the model upon receiving new data sample without making decision for the newly observed data sample. In this case, the importance of adaptation is unclear.

- The paper does not present any theoretical results in form of a theorem or lemma to present the error bound. It would be useful if the paper presents the error bound in the paper and compares the error bound of the proposed algorithm with that of the case where all samples in the buffer are employed to update the model.

**Summary Of The Paper:**

The paper studies the problem of online continual learning. The paper proposes a new algorithm which takes into account previously observed data samples instead of only the newly observed data sample.

**Summary Of The Review:**

In summary, the paper studies an important and interesting problem in online continual learning and proposes a new method for adaptivity of models in dynamic environments. However, the paper does motivate the importance of the proposed method well and to show the effectiveness of the proposed algorithm more analytical discussions should be added to the paper.

---

> ### Author Response · Authors · 2022-11-10
> **Response to reviewer kCCH**
>
> We would like to greatly thank the reviewer for constructive suggestions.  Please find the detailed responses below.
>
> *Q1. The paper does not motivate the benefit of selecting the most similar data samples.*
>
> A1. We made the following modification to better motivate the benefit.
> 1) In section 4.2 of the revised version, we compare the proposed method with Full adaptation, which uses all samples from the buffer to adapt the model. Feedforward adaptation is better than full adaptation, which means adapting models with more samples is not effective, because online adaptation focuses on the ability to rapidly learn and adapt in the presence of non-stationarity instead of generalization ability like offline learning. Feedforward adaptation is better than feedback and random adaptation, which means the sample selection strategy based on sample similarity in feedforward adaptation is more critical than the time-based sample selection and random sampling.
> 2) In this paper, we focus on prediction problems. Online decision-making is not covered by this work.  We believe that online adaptation for prediction problems is still important and valuable. It has applications in human trajectory prediction of human-robot collaboration [1], vehicle trajectory prediction, and traffic prediction  [2].
>
> *Q2. The paper does not present any theoretical results in form of a theorem or lemma.*
>
> A2. Thank you for the suggestions. In the revised version, we prove the error bound of the general online adaptation method as Theorem 1. We conclude the error bound for feedback and feedforward adaptation under a linear time-variant system as Lemma 1. And we conclude the error bound for feedback and feedforward adaptation under a time-invariant polynomial system as Lemma 2.
>
> Please let me know if you need further details.
>
> [1] Y. Cheng, et al., "Towards Efficient Human-Robot Collaboration With Robust Plan Recognition and Trajectory Prediction," in IEEE Robotics and Automation Letters, 2020.
> [2] Anastasios D. Doulamis, et al., "An adaptable neural-network model for recursive nonlinear traffic prediction and modeling of MPEG video sources." IEEE Transactions on Neural Networks, 2003.

---

> ### Comment · Reviewer_kCCH · 2022-11-15
> **Rebuttal Discussion**
>
> Thanks authors for the response. However, the authors' responses and revisions do not address my main concerns and I keep my score unchanged. I believe the adaptation is important when the agent receives data samples and makes prediction for the newly observed data samples in an online fashion. Therefore, in these cases theoretical analysis in terms of offline error does not properly show the adaptation ability of the algorithm. Furthermore, It would be beneficial for the paper that it compares the error bounds of feedforward adaptation with full adaptation to prove the effectiveness of the proposed method. However, the paper lacks such comparisons and discussions. I hope this review helps the authors to improve the paper.

---

> > ### Author Response · Authors · 2022-11-16
> > **Response to the discussion**
> >
> > Thank you for the further discussion. We prove the error bound of full adaptation in Appendix C.2 of the updated submission.
> > 1) Given a linear time-variant function, the expected error bound of the feedforward adaptation is smaller than the full adaptation.
> > 2) Given a time-invariant polynomial function,  the expected error (not the bound) of the feedforward adaptation is smaller than the full adaptation.
> >
> > In the experiments, we also show that the proposed feedforward adaptation is better than the full adaptation. It is reasonable because online adaptation focuses on the ability to rapidly learn and adapt in the presence of non-stationarity instead of generalization ability like offline learning [1]. Thus, optimizing with a few critical samples is more efficient than optimizing with all samples.
> >
> > We look forward to your feedback.
> >
> > [1] W. Si, et al, "AGen: Adaptable Generative Prediction Networks for Autonomous Driving," IEEE Intelligent Vehicles Symposium (IV), 2019.

---

> ### Comment · Reviewer_kCCH · 2022-11-18
> **Post Rebuttal Thoughts**
>
> Thanks authors for their clarification. After reading author responses and taking look at the paper again, I believe that in order to analyze the performance of the proposed algorithm, regret analysis can better evaluate the adaptation ability of the proposed algorithm compared to current analysis in the paper. Cumulative regret is defended as the cumulative difference between the loss of the algorithm and that of the best in hindsight through time. Different types of regret can be considered for the analysis. There are some convex optimization algorithms that can be applied to solve the problem of interest in the paper. In fact, a convex model $\hat f$ can be chosen and then using the existing online convex optimization algorithms regret upper bound of order $\mathcal O(\sqrt T)$ can be guaranteed. However, from reading the paper, I think that the proposed algorithm may only guarantee regret of $\mathcal O(T)$. After discussion with authors I decide to stay with my vote.

---

> > ### Author Response · Authors · 2022-11-18
> > **Response to the regrent minimization methods**
> >
> > Thank you for the further discussion.
> >
> > The objective of the feedforward adaptation is more reasonable than the regret minimization of general online learning, for the following reasons.
> > 1. The objective of the proposed feedforward adaptation is closer to real-world applications than regret minimization.
> > Note that in general online learning, regret analysis (and minimization) is only used for the posterior error. That means, after receiving the data, we minimize the cumulative fitting error on the known input-label pairs.  Actually, feedback adaptation is analyzing and minimizing the cumulative regret, as shown in equation (4).  We've reported convex optimization algorithms (e.g. SGD and EKF) on the results of feedback adaptation in the experiments. Which is our baseline. However, in real-world applications, we consider the prediction (prior) error in the future, instead of the regret for the posterior error, as shown in equation (3). Since we do not know the labels in the future, so we cannot compute and minimize the prior error (or prior regret). In our proposed feedforward adaptation, we minimize the bound of the prior error to approximate the prior error minimization, as shown in equation (5). With a  bit of misuse of concepts, if we make an analogy to classical offline machine learning, regret analysis and minimization (e.g. feedback adaptation) is minimizing the loss on the train set, however, the objective of the feedforward adaptation is minimizing the bound of the loss on the test set.
> >
> > 2. Compare the regret of the feedforward adaptation is not fair.
> > Regret analysis is used to minimize the posterior fitting error. But in the proposed feedforward adaptation, we minimize the bound of prior prediction error. Prior error and posterior error are not comparable. Actually, we do not have a so-called convergence rate on prior errors. To the best knowledge of the authors, there does not exist any work that could guarantee convergence on the prior error for the time-varying function.  All existing convergence analyses are based on the posterior error. Although we do not have the convergence rate on prior errors, feedforward adaptation minimizes the bound of the prior errors.
> >
> > 3. In experiments, we show that the proposed objective is better than regret minimization since it is more practical. In other words, in real-world applications feedforward adaptation is better than feedback adaptation (of cumulative regret minimization).
> >
> > 4. Unlike general online learning (with cumulative regret minimization) which learns a function that averagely performs better in a wide horizon of the past, our feedforward adaptation optimizes the function to perform the best in the next step. At every step, the process repeats and repeats to achieve the best performance in the next step. Which has some similarities with Model Predictive Control.

---

### Official Review · Reviewer_CSL7 · 2022-10-30

**Confidence:** 2
**Correctness:** 3
**Technical Novelty And Significance:** 2
**Empirical Novelty And Significance:** 2
**Recommendation:** 3

**Clarity, Quality, Novelty And Reproducibility:**

This paper is well-written and easy to follow in general. However, a clarity issue in Algorithm 1 is that $s$ in line 4 is not defined. The authors might want to add a clarification that $s$ is chosen based on some rules.

**Strength And Weaknesses:**

Strength:

This paper studies an interesting question that is impactful for the fields of online learning and online control.  The intuition behind the proposed change to existing online adaptation methods is convincing, and some intuitions have been verified in the first synthetic experiment. Specifically, I like the plot in Figure 1 (a) that shows how the performance of the proposed approach depends on the time-varying shift of the function.

Weakness:

I feel the major weakness of this work is on the theoretical part. If I understand correctly, the error bound in equation (8) is just an upper bound of the prediction error $e_{t+1}$. I don’t think one can claim feedforward is better than feedback because the $E[B_e^{ff}] \leq E[B_e^{fb}]$, because such claim can only be made when an upper bound is smaller than a lower bound. And it is also worth noticing that the proof only works for a specific special case. These two factors make me feel the theoretical guarantee is very weak.

Another weakness of this work is on the algorithm design. It has been shown both from the theory and the simulation that the proposed feedforward approach is prone to $\delta$, which characterize how fast the ground-truth prediction function changes. In other words, one needs to decide whether to use the feedforward approach based on $\delta$. However, the value of $\delta$ is not known before the online prediction process starts. Even if one can switch algorithms in the middle based on the historical experience, it is unclear how to estimate $\delta$ because the ground-truth prediction functions are unknown.

For the experiments, I’m uncertain about whether the improvement made on the real-world datasets are significant.

**Summary Of The Paper:**

This paper studies the problem of using online adaptation method in time-series prediction tasks. Existing approaches that feedback based on the latest prediction errors have the risk of forgetting past information. To address this challenge, the authors of this paper propose an approach that uses the critical data samples from the memory buffer to optimize the prediction model. The authors derive a bound of the prediction error, which gives the insight that the proposed approach will perform well if the ground-truth prediction functions are time-varying at a very slow rate. They also use experiments to show the proposed method performs better than existing approaches with and without adaptation.

**Summary Of The Review:**

In summary, I feel the contributions made in this work is not significant enough for a conference publication. The theoretical contributions are not very meaningful, and the novelty of algorithm design is limited. I'm uncertain about how significant the empirical improvements on the real-world datasets are. Therefore, I would vote for reject with a low confidence score.

---

> ### Author Response · Authors · 2022-11-10
> **Response to reviewer CSL7**
>
> We would like to thank the reviewer for your comments. Please find the detailed responses below.
>
> *Q1. The major weakness of this work is the theoretical part.*
>
> A1.  Minimizing the upper bound of the objective (e.g. prediction error) is reasonable and common if we cannot directly minimize the objective itself. Although our proof works for a special case, the case is very common.
> 1. The original goal of the adaptation is to minimize the prediction error in the future.  Due to the lack of future ground-truth values, it is not feasible to measure the prediction error. In that case, minimizing the upper bound of the future prediction error is a reasonable and natural solution [1], like we do in the proposed feedforward adaptation. (Lower bound is zero for all methods). In addition, optimizing the upper bound is equivalent to optimizing the worst-case scenario. In real-world applications, the worst-case analysis is very useful [2].  In our paper, Lemma 1 shows that, in a slow-varying system, the proposed method has a lower error bound. This is an important theoretical result because we cannot derive real error for unknown general functions.  Furtherly, when the system is a time-invariant polynomial, as shown in Lemma 2, the expectation of the prediction error (not a bound) of the proposed method is lower. At least for the time-invariant polynomial system, feedforward is probably better than feedback.  In addition, the experimental results on real-world data show that the real prediction error of the proposed method is lower than the baselines.
> 2.  We acknowledge that our proof works for a special case, but the case is very common. Note that, we only assume the system (or function) is slow-varying within recent L steps (e.g. L=100), not slow-varying for every step. So the assumption is only mild. In the experiments, we use four widely-used datasets from different domains to evaluate our algorithms. The results show that, for all four datasets, the proposed methods achieve the best performance.
>
> *Q2. The weakness of the algorithm is that rely on $\delta $.*
>
> A2. The adaptation algorithm does not rely on $\delta $. As shown in eq (12,13) and algorithm 2, the proposed feedforward adaptation method selects the most similar samples to the current observation.  Then using the selected samples to optimize the prediction model by Gradient Descent. The process does not contain $\delta $. In the uncertainty estimation,  is an approximated hyperparameter. In the experiments, we simply set  $\delta =0$, then still obtain affordable uncertainty estimation as shown in appendix D.6. The approximation accuracy of  $\delta $  does not affect prediction performance.
>
> *Q3.  I’m uncertain about whether the improvement made on the real-world datasets are significant.*
>
> A3. In real-world experiments, compared with the previous feedback method, we significantly improve the adaptation performance (MSE) by 12.6%, 8.0%, 22.3%, and 8.1% in all four datasets and two different models.
>
> Please let me know if you need further details.
>
>
> [1]  Jagdish S. Rustagi, ”Optimization techniques in statistics”.  Elsevier, 2014.
> [2] Tim Roughgarden. ”Beyond the worst-case analysis of algorithms”. Cambridge University Press, 2021

---

### Author Response · Authors · 2022-11-10
**Revision of the submission**

We thank all reviewers for their constructive comments and suggestions. We have revised the manuscript, and the significant changes are marked in red. The major modifications are summarized below.

1. In section 2.2, 3.2, and 4.2, we make the motivation clearer. Train-test distribution mismatch is very common in time-series prediction. Thus, online test-time adaptation is crucial. And our proposed feedforward adaptation is the most effective method in many cases.
2. In section 3.1 and 3.4, we conclude our theoretical contributions as Theorem 1 (Error Bound of Online Adaptation), Lemma 1 (Linear time-variant function), and Lemma 2 (Time-invariant polynomial function).
3. In section 4.2, we compare the proposed method with the new baseline: Full Adaptation which uses all samples to adapt the model. And the proposed method still outperforms the new baseline.
4. In section 4.2, we conduct a study about the sample selection strategy of different adaptation methods. The study shows that the sample selection strategy in the feedforward adaptation method could intrinsically extract the periodicity of the input data, while other methods cannot.
5. In appendix D.6, we provide the uncertainty estimation of the proposed method. Uncertainty estimation is another advantage of the proposed method beyond accurate prediction.

---

> ### Author Response · Authors · 2022-11-16
> **Update the revision**
>
> 6. In appendix C.2, we prove the error bound for full adaptation and random adaptation. As the result,  the expected error bound of the feedforward adaptation is smaller than the full adaptation and random adaptation.
>
>
> In conclusion, we theoretically prove that:
> (1) With a proper buffer size, feedforward adaptation always has a smaller error bound than full adaptation and random adaptation.
> (2) For a slow time-varying system, with the proper buffer size, feedforward adaptation achieves a smaller error bound than feedback adaptation.
> In the experiments, we show the effectiveness of the proposed feedforward adaptation method.

---

### Decision · Program_Chairs · 2023-01-20

**Decision:**

Reject

**Justification For Why Not Higher Score:**

Not so convincing in terms of theory and algorithm

**Justification For Why Not Lower Score:**

N/A

**Metareview: Summary, Strengths And Weaknesses:**

Authors have investigated to use online adaptation method in time-series prediction tasks. To address this challenge that catastrophic forgetting occurred in existing methods, it proposes a method to leverage critical data samples from the memory buffer to optimize the prediction model. They have conducted experiments to demonstrate that show the proposed method is effective.

The theoretical analysis is not solid and algorithm design has a flaw.


**Summary Of Ac-Reviewer Meeting:**

No need due to majority of unfavorable reviews